# On Uncertainty, Tempering, and Data Augmentation in Bayesian Classification

**Sanyam Kapoor**[*]
New York University

**Wesley J. Maddox**[*]
New York University

**Pavel Izmailov**[*]
New York University

**Andrew Gordon Wilson**
New York University

## Abstract

Aleatoric uncertainty captures the inherent randomness of the data, such as measurement noise. In Bayesian regression, we often use a Gaussian observation model, where we control the level of aleatoric uncertainty with a noise variance parameter. By contrast, for Bayesian classification we use a categorical distribution with no mechanism to represent our beliefs about aleatoric uncertainty. Our work shows that explicitly accounting for aleatoric uncertainty significantly improves the performance of Bayesian neural networks. We note that many standard benchmarks, such as CIFAR-10, have essentially no aleatoric uncertainty. Moreover, we show that data augmentation in approximate inference softens the likelihood, leading to underconfidence and misrepresenting our beliefs about aleatoric uncertainty. Accordingly, we find that a cold posterior, tempered by a power greater than one, often more honestly reflects our beliefs about aleatoric uncertainty than no tempering — providing an explicit link between data augmentation and cold posteriors. We further show that we can match or exceed the performance of posterior tempering by using a Dirichlet observation model, where we explicitly control the level of aleatoric uncertainty, without any need for tempering.

## 1 Introduction

Uncertainty is often compartmentalized into *epistemic* and *aleatoric uncertainty* [27, 33, 41]. Epistemic uncertainty, sometimes called model uncertainty, is the reducible uncertainty about which solution is correct given limited information. Bayesian methods naturally represent epistemic uncertainty through a distribution over model parameters, leading to a posterior distribution over functions that are consistent with data. As we observe more data, this posterior concentrates around a single solution. Aleatoric uncertainty is the irreducible uncertainty in data, often representing measurement noise in regression, or mislabeled training points in classification [e.g. 4]. Although measurement noise can be reduced, for example, with better instrumentation, it is often a fixed property of the data we are given. Correctly expressing our assumptions about aleatoric uncertainty is crucial for achieving good predictive performance, both with Bayesian and non-Bayesian models [56].

In particular, our assumptions about aleatoric uncertainty profoundly affect predictive *accuracy*, not only predictive uncertainty.[1] In Figure 1(a,b) we show the predictive distributions of two Gaussian process (GP) regression models [54] trained on the same data and using the same RBF kernel function. The only difference between these models is their assumptions about the aleatoric uncertainty. The model in Figure 1(a) assumes a high observation noise (each point is corrupted by $\mathcal{N}(0, \sigma^2)$,

---

[*]Equal contribution.

[1]Epistemic uncertainty also has a significant effect on predictive accuracy, see Wilson and Izmailov [62].

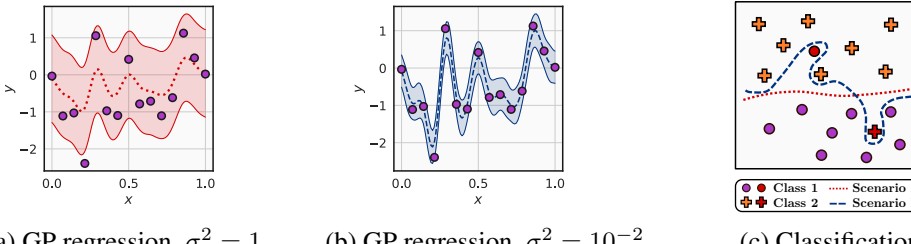

(a) GP regression, $\sigma^2 = 1$  (b) GP regression, $\sigma^2 = 10^{-2}$  (c) Classification

Figure 1: **Effect of aleatoric uncertainty.** In regression, we can express our assumptions about aleatoric uncertainty by setting the observation noise parameter $\sigma^2$. **(a)**: GP regression with high observation noise $\sigma^2$ explains many of the observations with noise. The datapoints are shown with purple circles, the dashed line shows the predictive mean, and the shaded region shows one standard deviation of the predictive distribution. **(b)**: GP regression on the same data, but with low observation noise $\sigma^2$ fits the training data nearly perfectly. **(c)**: In classification, we do not have a direct way to specify our assumptions about aleatoric uncertainty. We might use the same Bayesian neural network model if we know the data contains label noise (**Scenario A**) and if we know that there is no label noise (**Scenario B**), leading to poor performance in at least one of these scenarios.

with $\sigma^2 = 1$). Consequently, this model explains many of the observations with noise — such that the predictive mean does not closely fit the training datapoints. The model in Figure 1(b) assumes a low observation noise ($\sigma^2 = 10^{-2}$), and consequently the predictive mean runs through the training data, leading to very different predictions compared to the model in Figure 1(a).

Our assumptions about the aleatoric uncertainty are just as important in classification. We illustrate this point in Figure 1(c), where we fix a dataset and consider two possible scenarios. **Scenario A**: suppose we know that the data contains label noise, with some incorrectly labeled training examples. In this case, it is reasonable to believe that the data points shown in red in Figure 1(c) are highly likely to be incorrectly labeled, and we should opt for a solution with a decision boundary shown with a dotted red line. **Scenario B**: now, suppose that we know that the data are correctly labeled. In this case, we want our model to perfectly classify all the training data and would prefer a solution shown by the dashed blue line. Both of these solutions are reasonable descriptions of the data, and we can only make an informed choice between them by incorporating our assumptions about the aleatoric uncertainty. While in regression problems we can easily express our assumptions about aleatoric uncertainty, such as through a noise variance parameter in a Gaussian likelihood, in classification we do not have a direct way to express these beliefs. Indeed, in practice we might use the same Bayesian neural network (BNN) with the same likelihood and prior in both scenarios A and B above, necessarily leading to poor performance in at least one of these scenarios.

In this paper, we explicitly characterize aleatoric uncertainty in BNN classification. We show that *cold posteriors* [61] more honestly reflect our beliefs about aleatoric uncertainty than standard Bayesian posteriors. Further building on our analysis of aleatoric uncertainty in Bayesian classification, we describe an alternative to cold posteriors called the *noisy Dirichlet model*. The noisy Dirichlet model can be viewed as simply modifying the prior over the parameters of a BNN to put more mass on the solutions that make *highly confident* predictions on the training data, but only in scenarios with low aleatoric uncertainty. Unlike standard BNN classifiers, we get control over the aleatoric uncertainty through the prior. This approach achieves competitive performance on image classification problems with a valid Bayesian model and *removes the cold posterior effect*.

Finally, we show theoretically how data augmentation counterintuitively softens the likelihood, leading to solutions that underfit the training data. These results precisely characterize the empirical link between data augmentation and the success of cold posteriors observed by Izmailov et al. [31] and Fortuin et al. [14], and show that cold posteriors in fact serve to *correct* for the effects of data augmentation on aleatoric uncertainty.

## 2 Related Work

Early work on Bayesian neural networks focused on hyperparameter learning and mitigating over-fitting, using Laplace approximations, variational methods, and MCMC [e.g. 37, 25, 47]. More

recently, Wilson and Izmailov [62] show how Bayesian model averaging is especially compelling for *modern* deep neural networks. A number of recent works have focused on making Bayesian deep learning practical [e.g., 63, 38, 49, 65, 12, 11], with improved performance, and essentially no overhead.

**Modeling label noise.** In the context of label noise for classification, prior works focus on modifying the loss function. For example, Collier et al. [9, 10] propose input-dependent noise losses for label noise in classification to improve accuracy, but do not focus on the uncertainty estimates. Similarly, Zhang and Sabuncu [66] and Feng et al. [13] propose modifications of the standard cross-entropy loss to provide increased robustness to label noise, but do not focus on modeling the inherent uncertainty in label noise settings.

**Dirichlet-based uncertainty (DBU).** Recent literature proposes neural networks where the output is a Dirichlet distribution, capturing both the epistemic and aleatoric uncertainty [41, 57, 40, 42, 7, 67, 46, 59, 58, 34]. These models are generally proposed as efficient alternatives to ensembles and Bayesian models, and targeted for improved uncertainty quantification, anomaly detection and adversarial robustness. Joo et al. [32] reformulate the problem of minimizing the softmax cross-entropy loss over the data into a collection of distribution matching problems. In this paper, we study the Dirichlet distributions associated with the posterior over the predictive class probabilities, which arise naturally in standard Bayesian classification models. Contrary to the methods in the DBU family, we do not modify the observation model or the structure of the network.

**Cold posterior effect.** Wenzel et al. [61] demonstrated with several examples that raising the posterior to a power $1/T$, with $T < 1$, in conjunction with SGLD inference, improves performance over classical training, but $T = 1$ can provide worse results than classical training, in what has been termed the *cold posterior effect*. However, in a study using HMC inference, Izmailov et al. [31] showed that for *all* cases considered in Wenzel et al. [61] there is no cold posterior effect when data augmentation is removed. The sufficiency of data augmentation to observe the cold posterior effect has been confirmed by several works using SG-MCMC inference [14, 48, 45]. Several works have suggested that misspecified priors explain cold posteriors, arguing that the current default choice of isotropic Gaussian priors in BNNs are inadequate [61, 64, 14]. However, Fortuin et al. [14] find that the cold posterior effect cannot be alleviated for convolutional neural networks by using heavy tailed or correlated priors under data augmentation. With HMC inference and no data augmentation, Izmailov et al. [31] also show that standard Gaussian priors perform similarly to a range of other priors, such as heavy-tailed logistic or mixtures of Gaussian priors, and generally are high performing. Noci et al. [48] argue that many different factors — likelihoods, priors, data augmentation — can cause cold posteriors, and that there may be no one cause in general. Aitchison [2] argue that BNN likelihoods are misspecified since many benchmark datasets have been carefully curated by human labelers. They also show that posterior tempering is less helpful in the presence of label noise. Adlam et al. [1] also consider model misspecification, showing a cold posterior effect can arise even with exact inference in GP regression when the aleatoric uncertainty is misestimated. Nabarro et al. [45] modify the likelihood to accommodate data augmentation, using as the observation model the average of the neural network outputs over augmentations, but still find a cold posterior effect. In concurrent work, Bachmann et al. [3] provide an alternative argument to how data augmentations introduce misspecification due to correlation between errors which cold posteriors correct for.

Our paper makes several distinctive contributions in the context of prior work: (1) we argue that standard likelihoods do not represent our beliefs about aleatoric uncertainty, and that standard benchmarks have essentially no aleatoric uncertainty; (2) we show how tempering and data augmentation specify aleatoric uncertainty, beyond curation; (3) we show the *precise way in which data augmentation with SGLD leads to underconfidence in the likelihood*, a counterintuitive result which finally resolves the empirical connection between data augmentation and cold posteriors; (4) we show that a $T < 1$, particularly with data augmentation, is a more honest reflection of our beliefs about aleatoric uncertainty than $T = 1$; (5) we show how the noisy Dirichlet likelihood, originally used for tractable GP classification [43], can naturally reflect our beliefs about aleatoric uncertainty, and for the first time remove the cold posterior effect in the presence of data augmentation.

## 3 Background

**Bayesian Model Averaging.** With Bayesian inference, we aim to infer the posterior distribution over parameters having observed a dataset $\mathcal{D} = \{(x_i, y_i)\}_{i=1}^N$ of input-output pairs, given by $p(w \mid \mathcal{D}) \propto p(\mathcal{D} \mid w)p(w)$ for a given observation likelihood under the i.i.d. assumption $p(\mathcal{D} \mid w) = \prod_{(x,y) \in \mathcal{D}} p(y \mid x, w)$, and prior over parameters $p(w)$. For any novel input $x_\star$, we compute the posterior predictive distribution via *Bayesian model averaging* (BMA) given by

$$p(y_\star \mid x_\star) = \int p(y_\star \mid x_\star, w)p(w \mid \mathcal{D})dw. \tag{1}$$

The BMA integral in Eq. (1) cannot generally be expressed in closed form for a Bayesian neural network, and is typically approximated with variational inference (VI), the Laplace approximation, or Markov Chain Monte Carlo (MCMC) [e.g., 62, 44].

**Bayesian Classification.** For a $C$-class classification problem, a standard choice of observation likelihood is the categorical distribution $p(y \mid x, w) = \mathrm{Cat}([\pi_1, \ldots, \pi_C])$, where each class probability is computed using a softmax transform of logits $g(x; w) = [g_1(x; w), \ldots, g_C(x; w)] \in \mathbb{R}^C$ as $\pi_c \propto f_c(x; w) = \exp\{g_c(x; w)\}$, and hence called the *softmax likelihood*. The prior over parameters $p(w)$ is often chosen to be an isotropic Gaussian $\mathcal{N}\left(0, \sigma^2 I\right)$ with some fixed variance $\sigma^2$. Approximate posterior samples are often obtained via SGLD [60], and used for a Monte Carlo estimator of the BMA in Eq. (1).

**Cold Posteriors.** Let $U(w) = -\log p(\mathcal{D} \mid w) - \log p(w)$ denote the posterior energy function of a BNN. Following Wenzel et al. [61], we define a *cold posterior* for $T < 1$ as

$$p_{\mathrm{cold}}(w \mid \mathcal{D}) \propto \exp\left\{-\frac{1}{T}U(w)\right\}, \tag{2}$$

which effectively raises both the likelihood and the prior to a power $1/T$. $T = 1$ recovers the standard Bayes' posterior. A *tempered likelihood posterior* (e.g. [17]) only raises the likelihood to a power $1/T$ as

$$p_{\mathrm{temp}}(w \mid \mathcal{D}) \propto p(\mathcal{D} \mid w)^{1/T}p(w). \tag{3}$$

## 4 Aleatoric Uncertainty in Bayesian Classification

We will now discuss how we represent aleatoric uncertainty in classification (Section 4.1), how tempering reduces aleatoric uncertainty (Section 4.2), and how we can explicitly represent aleatoric uncertainty without tempering with a noisy Dirichlet observation model (Section 4.3).

### 4.1 How do we represent aleatoric uncertainty in classification?

Let us consider the Bayesian neural network posterior over parameters $w$ in a classification problem: $p(w \mid D) \propto p(w) \prod_{(x,y) \in \mathcal{D}} f_y(x; w)$, where we denote the output of the softmax layer of the model corresponding to class $y$ on input $x$ as $f_y(x; w)$.

Alternatively, we can think of the class probability vectors $f(x) \triangleq f(x; w) = [f_1(x; w), \ldots, f_C(x; w)]$ as latent variables[2] in a $C - 1$ simplex $\Delta^{C-1}$, such that $\sum_{c=1}^C f_c(x; w) = 1$. A prior distribution $p(w)$ over the parameters of the network implies a joint prior distribution over $f = \{f(x)\}_{x \in \mathcal{D}}$. We can then do inference directly in *function space* over the variables $f$, yielding a posterior $p(f \mid \mathcal{D})$ equal to the distribution over $f$ implied by the the parameter space posterior $p(w \mid \mathcal{D})$ [see e.g. 54, Ch. 2.2]. We will use this function space posterior to analyze the assumptions on aleatoric uncertainty encoded by the Bayesian classification models.

Let us consider the likelihood $p(y \mid f(x))$ for a single observation $(x, y)$:

$$\begin{aligned} p(y \mid f(x)) = f_y(x) &\propto_{f(x)} C! \cdot f_y(x) \\ &= \mathrm{Dir.}(1, \ldots, 1, \underbrace{2}_{\text{position } y}, 1, \ldots, 1)(f(x)), \end{aligned} \tag{4}$$

---

[2] We use $f(x; w)$ and $f(x)$ interchangeably to denote the latent variable corresponding to the class probability vector on the input $x$. We use $f(x; w)$ to emphasize the dependence on the weights $w$.

where the left hand side is equal to the right hand side of Eq. (4) up to a scalar multiplier $C!$ (normalizing constant of the Dirichlet distribution), which does not depend on $f(x)$.[3] We note that the algebra in Eq. (4) is the same as the algebra used to show the conjugacy of the Dirichlet prior and the categorical likelihood [e.g. 6, Ch. 2.2]. Eq. (4) describes the distribution induced by the observation $y$ on the predicted class probabilities for the corresponding input $x$ by the likelihood. Consequently, the joint posterior over the class probability vectors $f = \{f(x)\}_{x \in \mathcal{D}}$ with the full dataset can then be written as

$$p(f \mid \mathcal{D}) \propto_f p(f) \prod_{(x,y) \in \mathcal{D}} \mathrm{Dir.}(1, \ldots, 1, \underbrace{2}_{\text{position } y}, 1, \ldots, 1)(f(x)). \tag{5}$$

Eq. (5) provides intuition for how BNNs estimate aleatoric uncertainty in classification. If we further assume that the implied prior $p(f)$ over $f(x)$ for each $x$ is uniform on $\Delta^{C-1}$ and independent across different inputs $x$, then for an observation $(x, y)$ the posterior over $f(x)$ is $\mathrm{Dir.}(1, \ldots, 2, \ldots, 1)$ and the posterior mean for the probability of the correct class $y$ is $\mathbb{E}[f_y(x)] = 2/(C + 1)$. For example, with a dataset containing 100 classes (e.g. CIFAR-100), the model on average will only be less than 2% confident in the correct label on the *training data*!

In practice, however, the prior $p(w)$ will imply a non-trivial joint distribution $p(f)$ over $\{f(x; w)\}_{x \in \mathcal{D}}$, so the actual posterior may be more (or less) confident than suggested by the analysis above. Moreover, we have a very limited understanding of the implied distribution $p(f)$ [see 61, 62, for empirical analysis]. In particular, most practitioners use simple $p(w) = \mathcal{N}(0, \alpha^2 I)$ priors, regardless of the amount of label noise in the data. Explicitly constructing a prior in the parameter space that leads to highly confident predictive distributions over $f(x)$ is challenging, but we will show how modifying the likelihood can be viewed as providing an input-dependent prior that more accurately reflects our beliefs about aleatoric uncertainty in Section 4.3.

Eq. (5) suggests two natural ways of modifying the posterior to account for the aleatoric uncertainty: (i) increasing the Dirichlet concentration for the observed class (equivalent to likelihood tempering), or (ii) decreasing the concentration for the unobserved classes (noisy Dirichlet model). Next, we describe both of these approaches in detail.

## 4.2 Likelihood tempering reduces aleatoric uncertainty

The tempered likelihood posterior over the parameters $w$ in Eq. (3) for BNN classification can be written as $p_{\text{temp}}(w \mid \mathcal{D}) \propto_w p(w) \prod_{(x,y) \in \mathcal{D}} f_y(x; w)^{1/T}$, where $T$ is the temperature. Analogous to the derivation above, we can rewrite the corresponding function space posterior as

$$p_{\text{temp}}(f \mid \mathcal{D}) \propto_f p(f) \prod_{(x,y) \in \mathcal{D}} \mathrm{Dir.}\big(1, \ldots, 1, \underbrace{1 + 1/T}_{\text{position } y}, 1, \ldots, 1\big)(f(x)), \tag{6}$$

In other words, the tempered posterior corresponds to the same model as the regular posterior, but assumes that we observed $1/T$ counts of class $y$ for each input $x$ in the training data. Assuming the prior $p(w)$ implies a uniform distribution over $f(x; w)$, the confidence in the correct labels on the train data under the tempered posterior $\mathbb{E}[f_y(x)] = (1 + 1/T)/(C + 1/T)$. For a dataset with 100 classes, at $T = 10^{-2}$ we get a much higher average confidence of more than 50% on the training data, much higher than the 2% for the standard observation model.

**Tempering the Likelihood vs Posterior.** Prior work has mostly considered tempering the full Bayesian posterior as opposed to just the likelihood (see Section 3). In Appendix G.2 we show that tempering the full Bayesian posterior is almost always equivalent to changing the prior distribution, and tempering the likelihood, and that likelihood tempering recovers the cold posterior effect.

**How should we think about tempering?** Prior work has asserted that posterior tempering sharply deviates from the Bayesian paradigm and the cold posterior effect is highly problematic [61, 48, 14]. We, on the other hand, argue that likelihood tempering is in fact a practical way to incorporate our assumptions about aleatoric uncertainty. Relatedly, Aitchison [2] argue that the cold posterior effect can be caused by data curation. For example, CIFAR-10 and ImageNet datasets have relatively very little label noise and are carefully curated [e.g. 53]. Thus, we may expect that tempered likelihoods with low temperatures will lead to optimal performance.

---

[3]We use notation $\propto_\theta$ to denote proportionality with respect to $\theta$.

**Is a Tempered Likelihood a Valid Likelihood?** The observation model is a distribution $p(y \mid x)$ over the labels conditioned on the input. The tempered softmax likelihood is in general not a valid likelihood because it does not sum to 1 over classes. But Wenzel et al. [61] argue that the tempered softmax likelihood with $T < 1$ can be interpreted as a valid likelihood if we introduce a new class which is not observed in the training data. Moreover, from Eq. (6), we can naturally interpret the tempered likelihood as using the multinomial observation model, assuming $1/T$ counts of the label are observed for each of the training datapoints, which is uncontroversial and perfectly valid.

**Smoothed softmax model.** Guo et al. [19] show that softmax temperature scaling can be used to fix the uncertainty calibration of trained neural networks. Analogously, we can simply divide the logits of the model by a scalar temperature $T$, before computing the softmax likelihood, resulting in the *smoothed softmax model*, a valid Bayesian model with a tunable temperature parameter. In Appendix F, we show that the smoothed softmax behaves differently than the tempered softmax likelihood, and is insufficient to model the aleatoric uncertainty in Bayesian classification. In particular, Figure 10 shows that the smoothed softmax likelihood does not address the cold posterior effect.

### 4.3 Noisy Dirichlet model: changing the prior over class probabilities

As we have seen in Eq. (6), likelihood tempering increases the posterior confidence by increasing the Dirichlet concentration for the observed class $y$ from 2 to $1 + 1/T$. We can achieve a similar effect by *decreasing the concentration of the unobserved classes*[4]. Indeed, consider the distribution

$$p_{\text{ND}}(f \mid \mathcal{D}) \propto_f \quad p(f) \prod_{(x,y) \in \mathcal{D}} \text{Dir.}(\alpha_\epsilon, \ldots, \alpha_\epsilon, \underbrace{\alpha_\epsilon + 1}_{\text{position } y}, \alpha_\epsilon, \ldots, \alpha_\epsilon)(f(x)), \tag{7}$$

where ND in $p_{\text{ND}}$ stands for *noisy Dirichlet* and $\alpha_\epsilon$ is a tunable parameter. The noisy Dirichlet model was originally proposed by Milios et al. [43] in the context of Gaussian process classification, where they designed a tractable approximation to this model. Using our running example, if $p(w)$ induces a uniform distribution over $f(x; w)$, for a problem with 100 classes and $\alpha_\epsilon = 10^{-2}$ we have expected confidence $\mathbb{E}[f_y(x)] = (\alpha_\epsilon + 1)/(C\alpha_\epsilon + 1)$, slightly more than 50%, similar to the tempered likelihood with $T = 10^{-2}$.

Going back to the parameter space, we can write the posterior distribution over $w$ corresponding to the noisy Dirichlet model as

$$p_{\text{ND}}(w \mid \mathcal{D}) \propto_w \quad p(w) \prod_{(x,y) \in \mathcal{D}} \text{Dir.}(\alpha_\epsilon, \ldots, \alpha_\epsilon, \underbrace{\alpha_\epsilon + 1}_{\text{position } y}, \alpha_\epsilon, \ldots, \alpha_\epsilon)(f(x; w)). \tag{8}$$

**The noisy Dirichlet model corresponds to a different prior.** Using the conjugacy of the Dirichlet and multinomial distributions [e.g., 6, Ch. 2.2.1], we can rewrite $p_{\text{ND}}$ from Eq. (8) as follows:

$$p_{\text{ND}}(w \mid \mathcal{D}) \propto_w \quad p(w) \prod_{(x,y) \in \mathcal{D}} \text{Dir.}(\alpha_\epsilon, \ldots, \alpha_\epsilon)(f(x; w)) \cdot f_y(x; w) =$$
$$q_{\text{ND}}(w) \prod_{(x,y) \in \mathcal{D}} f_y(x; w). \tag{9}$$

We can interpret $q_{\text{ND}}(w) \triangleq p(w) \cdot \prod_{x \in \mathcal{D}} \text{Dir.}(\alpha_\epsilon, \ldots, \alpha_\epsilon)(f(x; w))$ as a prior over the parameters $w$ of the model. Indeed, $q_{ND}(w)$ does not depend on the labels $y$, and simply forces the predicted class probabilities to be confident in any one of the classes for the training data. Then, the only difference between the noisy Dirichlet model in Eq. (9) and the standard Bayesian neural network for classification is the choice of the prior over the parameters: instead of using a generic prior $p(w)$ such as the standard Gaussian distribution, we use the noisy Dirichlet prior $q_{ND}$. See the *EmpCov* prior in Izmailov et al. [30] for another example of a prior that depends on the training inputs but not training labels.

The noisy Dirichlet prior is intuitively appealing: in many practical settings a priori we believe that the aleatoric uncertainty on the training data is low, and the model should be confident in one of the

---

[4]While we discuss increasing the confidence of the model here, we can equivalently decrease the confidence of the model by *increasing* the concentration parameters for the unobserved classes.

classes. At the same time, we would not want to enforce low aleatoric uncertainty everywhere in the input space, as we do not expect that all the possible inputs to the model should be classified with high confidence.

**The noisy Dirichlet model is a valid Bayesian model.** In short, the noisy Dirichlet model corresponds to using a *valid likelihood* — the softmax likelihood — with a prior that explicitly enforces the model predictions to be confident on the training data. In Section 6.2 we will see that the noisy Dirichlet model removes the cold posterior effect.

**Gaussian approximation.** Milios et al. [43] considered the distribution in Eq. (9) in the context of Gaussian process classification, but with a different goal: they aimed to create a regression likelihood which would approximate the softmax likelihood. We describe their Gaussian approximation which amounts to solving a regression problem in the space of logits $z(x)$ with a Gaussian observation model in Appendix D. While theoretically we do not need to use the Gaussian approximation for Bayesian neural networks and can directly use Eq. (9), we found that in practice the approximation is much more stable numerically. We use the Gaussian approximation in all experiments.

## 5 The Effect of Data Augmentation

Data augmentation is a key ingredient of modern deep learning. Surprisingly, however, data augmentation has not been studied in the context of Bayesian methods until recently. Several works have shown that data augmentation often causes the cold posterior effect in BNNs [31, 64, 14, 45]. We show that data augmentation, as applied in practice[5], is closely connected to likelihood tempering with $T > 1$. In this case, the cold posteriors counterbalance the effect of data augmentation.

First, we consider the interaction between data augmentation and SGLD (or, equivalently, any other SGMCMC method). With SGLD, we aim to construct an unbiased estimator of the gradient of the posterior log-density (see Appendix C). In prior work, e.g. in Wenzel et al. [61], the stochastic gradient is estimated using a randomly augmented mini-batch of the data. For a minibatch of the full dataset $\mathcal{D}_m = \{x_i, y_i\}_{i=1}^m \subset \mathcal{D}$, and a finite set of augmentations $\mathcal{T} = \{t_1, \ldots, t_K\}$, this stochastic gradient is given by $\nabla \widetilde{U}(w) = \frac{N}{m} \sum_{(x_i, y_i) \in \mathcal{D}_m} \nabla_w \log p(y_i \mid t_j(x_i)) + \nabla_w \log p(w)$, where the transformations $t_j$ are sampled uniformly from $\mathcal{T}$. There are two sources of randomness in $\nabla \widetilde{U}(w)$ — the choice of the mini-batch $\mathcal{D}_m$ and the choice of augmentation $t_j$ used for each $x_i$. The limiting distribution of SGLD is determined by the expectation of the stochastic gradient above, given by

$$\sum_{i=1}^N \mathbb{E}_t[\nabla_w \log p(y_i | t(x_i))] + \nabla_w \log p(w) = \sum_{i=1}^N \sum_{j=1}^K [\nabla_w \log p(y_i | t_j(x_i))^{\frac{1}{K}}] + \nabla_w \log p(w), \tag{10}$$

where $t_j \sim \mathcal{T}$ are augmentations sampled uniformly from $\mathcal{T}$. Therefore, we conclude that the limiting distribution of SGLD under data augmentation is given by

$$p_{\text{aug}}(w \mid \mathcal{D}) \propto_w p(w) \prod_{i=1}^N \prod_{j=1}^K p(y_i \mid t_j(x_i))^{1/K}. \tag{11}$$

We can interpret the limiting distribution in Eq. (11) as a *tempered likelihood posterior* for a new dataset $\mathcal{D}' = \{(t_j(x_i), y_i)\}_{i,j}$ which contains all augmentations of every data point from the original dataset $\mathcal{D}$. Furthermore, the likelihood is tempered with a temperature $K > 1$, corresponding to a *warm posterior*. In other words, the limiting distribution of SGLD with data augmentation corresponds to a posterior over the augmented dataset with a *softened* likelihood, leading to less confidence about the labels. By applying a cold temperature $T = 1/K$ to the posterior in Eq. (11), we can recover the standard Bayesian posterior on the augmented dataset $\mathcal{D}'$. In Section 6.2, we explore the effect of data augmentation on aleatoric uncertainty in practice and show that it does indeed soften the likelihood and lead to increased aleatoric uncertainty.

---

[5]Data augmentation is a practical procedure, where at each iteration of optimization or sampling, we draw random augmentation for each object in each mini-batch of the data. Consequently, we can only reason about the effect of the data augmentation on the limiting distributions of approximate inference procedures: data augmentation is not defined for *true* Bayesian neural networks.

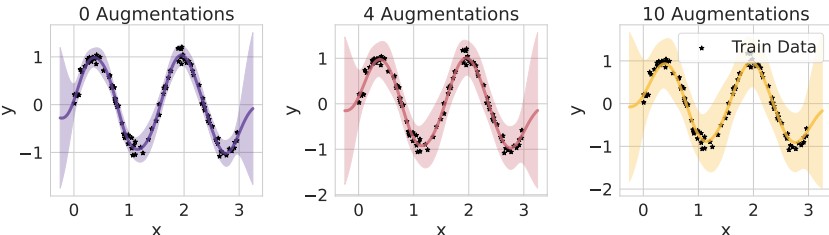

Figure 2: **Effect of data augmentation on GP regression.** As we increase the number $K$ of augmentations of the dataset, a GP model fit becomes more diffuse. Here the original training data is shown with black stars, and the augmented datapoints $t_j(x) = x + \tau_j$ are not shown. Both the predictive mean and confidence change with the number $K$ of augmentations.

**What about Other Inference Procedures?**   While the derivation in Eq. (11) comes directly from studying the stochastic gradient evaluated in SGMCMC, we can equivalently derive the same posterior distribution in stochastic variational inference [26] as we show in Appendix I.

**Should we always use temperature $T = 1/K$?**   While using the temperature $T = 1/K$ recovers the standard Bayesian posterior on the augmented dataset $\mathcal{D}'$, it is not necessarily a correct approach to modeling data augmentation. Indeed, the standard posterior $p(w \mid \mathcal{D}') \propto p(w) \prod_{i=1}^{N} \prod_{j=1}^{K} p(y_i \mid t_j(x_i))$ assumes independence across both augmentations and data points. In practice, however, we share the same label $y_i$ across all the augmented versions of the image; treating the observations as independent then leads to *underestimating* the aleatoric uncertainty. Indeed, consider the extreme scenario, where all the augmentations $t_j(x)$ simply return $x$. In this case, treating the observations $(t_j(x), y)$ as independent will simply raise the likelihood to a power of $K$, while in reality we have not received any additional information from the augmentation policy. Correcting the likelihood by a factor of $1/K$ is the correct approach only when the predictions on $t_j(x)$ are completely independent from each other (see Section 6.1). See also Nabarro et al. [45] for a related discussion.

# 6   Experiments

In this section we provide empirical support for the observations presented in Sections 4 and 5. First, in Section 6.1 we visualize the effect of data augmentation on the limiting distribution of SGMCMC using a Gaussian process regression model. Next, in Section 6.2 we report the results for BNNs on image classification problems. In Appendix E we additionally illustrate the effects of tempering, noisy Dirichlet model and data augmentation using a Bayesian neural network on a synthetic 2-D classification problem. Appendix K provides detailed numerical results for reference. All results report the mean and one standard deviation over 3 trials. The code to reproduce experiments is available at github.com/activatedgeek/understanding-bayesian-classification.

## 6.1   Visualizing the Effect of Data Augmentation

To illustrate our analysis in Section 5 we show the posterior in Eq. (11) under data augmentation in a Gaussian process (GP) regression model. We use a GP with a RBF kernel, noting that data points where $k(x, x') \approx 0$ are effectively independent from each other, which occurs when $\|x - x'\| \gg \delta$. Consequently, the implied correlations between inputs beyond the distance $\delta$ are zero. We use augmented samples that are given by $t_j(x) = x + \tau_j$, for vectors $\{\tau_j\}_{j=1}^{K}$ such that $\|\tau_j\| \gg \delta$, assuming that $k(x, t_j(x')) = 0$ for all datapoints $x, x'$ and all augmentations $t_j$. Thus, the outputs of the GP on the training data are independent from the outputs on the augmented datapoints.

We show the posterior near the original data in Eq. (11) for $K = 1, 4$ and 10 augmentations per datapoint in Figure 2. As the predictions on the train data are independent from the predictions on the augmented datapoints, the posterior in Eq. (11) corresponds to tempering the posterior on the original training data with a warm temperature equal to the number $K$ of augmentations. As a result, increasing $K$ softens the likelihood, leading to a less confident fit on the training data.

## 6.2 Image Classification with Bayesian Neural Networks

In this section we experimentally verify the results of the analysis in Sections 4 and 5 for Bayesian neural networks in image classification problems. First, we show that both the noisy Dirichlet model and tempered softmax likelihood can be successfully used to express our beliefs about the amount of label noise in the data. Then, we show that the noisy Dirichlet model does not require tempering to achieve optimal performance. Finally, we show that data augmentation softens the likelihood for BNNs, and that the optimal temperature depends on the complexity of the data augmentation policy. For all experiments we use a ResNet-18 model [20] and the CIFAR-10 [35] and Tiny Imagenet [36] datasets. We use the SGLD sampler with a cyclical learning rate schedule [60, 65] to sample from the posterior. We provide details on the hyper-parameters in Appendix L.

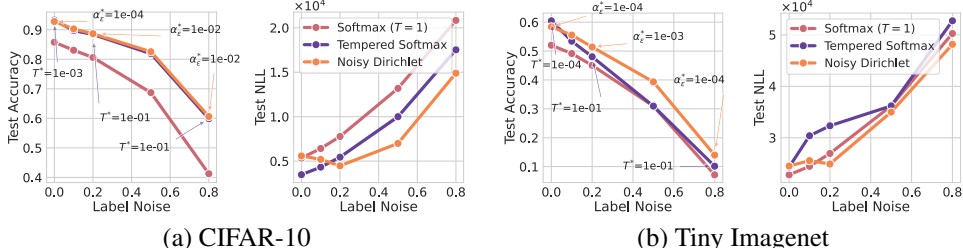

(a) CIFAR-10              (b) Tiny Imagenet

Figure 3: **Label noise in image classification.** BMA test accuracy and average sample NLL for the softmax, tempered softmax and noisy Dirichlet model **(a)** on CIFAR-10 and **(b)** Tiny Imagenet. Accounting for the label noise via the noisy Dirichlet model or the tempered softmax likelihood significantly improves accuracy accross the board. Moreover, the best performance is achieved by different values of temperature $T^*$ in the tempered softmax likelihood or noise $\alpha_\epsilon^*$ in the noisy Dirichlet model, i.e. no one model describes the data optimally across all levels of label noise.

**Modeling label noise.** In classification problems the aleatoric uncertainty corresponds to the amount of label noise in the data. Throughout the paper we argued that aleatoric uncertainty is misrepresented by standard Bayesian neural networks, and that likelihood tempering and the noisy Dirichlet model are compelling approaches to incorporate information about the amount of label noise. In Figure 3, we show the BMA test accuracy and average sample negative log-likelihood (NLL) for the standard softmax likelihood, tempered softmax likelihood and the noisy Dirichlet model on CIFAR-10 and Tiny Imagenet under varying amounts of label noise. We plot the results for the best performing temperature $T \in [10^{-5}, 10]$ or noise $\alpha_\epsilon \in [10^{-6}, 10^{-1}]$ for the noisy Dirichlet model. On both datasets and across label noises, we can significantly improve performance over the softmax likelihood by explicitly modeling the aleatoric uncertainty with either tempering or the noisy Dirichlet model.

**No cold posterior effect in the noisy Dirichlet model.** We explore the effect of posterior tempering on the standard softmax classification likelihood and the noisy Dirichlet model with noise parameter $\alpha_\epsilon = 10^{-6}$. In Figure 4 we show the results on the CIFAR-10 dataset. As reported by Wenzel et al. [61], for the standard softmax likelihood, tempering is required to achieve optimal performance, with $T = 10^{-3}$ providing the best results. For the noisy Dirichlet model on the other hand, tempering does not significantly improve the results with roughly constant performance across the temperature values. In particular, the noisy Dirichlet model achieves near-optimal results at $T = 1$. These results agree with our analysis in Section 4: both tempering and the noisy Dirichlet model are alternative ways of expressing our beliefs about the aleatoric uncertainty in the data; with the noisy Dirichlet model, we can achieve strong results without any need for tempering.

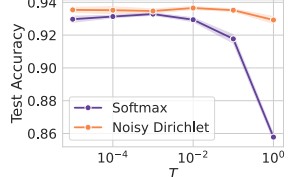

Figure 4: BMA test accuracy for the noisy Dirichlet model with $\alpha_\epsilon = 10^{-6}$ and the softmax likelihood as a function of temperature on CIFAR-10. The noisy Dirichlet model shows no cold posterior effect.

**Data augmentation leads to underfitting on train data.** In Figure 5a we show the average sample train NLL for the models trained with and without data augmentation. Across temperatures for the tempered softmax and $\alpha_\epsilon$ parameters for the noisy Dirichlet model, adding data augmentations reduces the quality of the

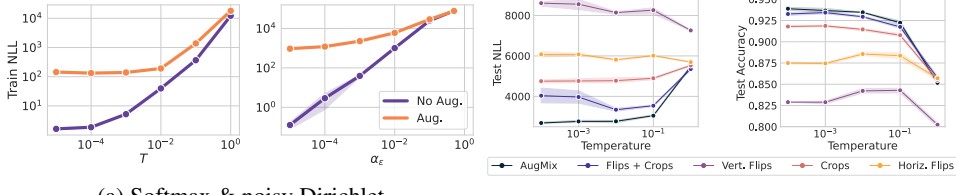

(a) Softmax & noisy Dirichlet

(b) Augmentations vs Tempering

Figure 5: **Effect of data augmentation on BNN image classification.** For all plots we use ResNet-18 on the CIFAR-10 dataset. **(a)**: NLL on the train data for the tempered softmax model across different temperatures $T$ and the noisy Dirichlet model across different values of noise $\alpha_\epsilon$, with and without data augmentation. As predicted in Section 5, data augmentation softens the likelihood, leading to a more diffuse fit on the train data. Tempering or reducing the noise parameter $\alpha_\epsilon$ is needed to counteract the effect of data augmentation. **(b)** BMA test accuracy and NLL for various augmentation policies. As predicted in Section 5, more complex policies corresponding to a higher effective number of augmentations $K$ require lower temperatures.

fit on the original training data. This result is analogous to the GP results in Section 6.1 and our analysis in Section 5: data augmentation softens the likelihood, leading to a more diffuse fit.

**Complex augmentations require lower temperatures.** In Section 5 we showed that under data augmentation the likelihood is effectively tempered with a temperature $K$, assuming the predictions on the augmented datapoints are completely independent from the original datapoints. In practice however, the augmentations can be close to the original images, and the corresponding predictions can be highly correlated. In Figure 5b, we consider five separate types of augmentations for ResNet-18 on CIFAR-10 at various posterior temperatures: horizontal and vertical flips, random crops, combinations of flips and crops, and AugMix [23] — an augmentation policy employing a very diverse set of transformations. We find that, as predicted by the analysis in Section 5, the optimal temperatures are different for different policies: in terms of test NLL, the simple policies (vertical and horizontal flips) corresponding to $K = 2$ work best at warmer temperatures $T = 1$, intermediate policies (crops and crops+flips) corresponding to $K \approx 100$ require lower temperatures $T \in [10^{-2}, 10^{-1}]$, and the most complex AugMix policy requires the lowest temperature $T \leq 10^{-4}$.

**Robustness to corruptions.** Finally, we consider robustness to dataset corruptions by considering the CIFAR-10C dataset [22], evaluating our models with both tempered softmax (TS) and noisy Dirichlet models. A full plot is shown in Appendix K; summarizing those results, we see that TS has the best performance at $T = 10^{-3}$ with an accuracy of 93.2 (with $T = 1$ again under-performing), while again the noisy Dirichlet model performs well, with an accuracy of 92.6 (at $\alpha_\epsilon = 10^{-4}$).

## 7   Discussion

A correct representation of aleatoric uncertainty is crucial to achieve strong performance with Bayesian neural networks. Standard Bayesian classifiers have no explicit mechanism to represent our beliefs about aleatoric uncertainty, which can lead to inadequate fits to the training data. This effect is exaggerated by data augmentation, which softens the likelihood, making the models under-confident on the training data. Posterior tempering is a simple and practical way to correct for this misspecification and express our beliefs about the aleatoric uncertainty: most benchmark datasets have very low levels of label noise, and we can express this belief by tempering the softmax likelihood. We also showed that we can achieve a similar effect without tempering, by using a prior that forces the model to be confident on the training datapoints with the noisy Dirichlet model.

## Acknowledgements

We would like to thank Micah Goldblum, Wanqian Yang, and anonymous referees for very helpful comments. This research is supported by an Amazon Research Award, NSF I-DISRE 193471, NIH R01DA048764-01A1, NSF IIS-1910266, and NSF 1922658 NRT-HDR: FUTURE Foundations, Translation, and Responsibility for Data Science. This work is supported in part through the NYU IT High Performance Computing resources, services, and staff expertise.

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
