# Appendix for
# On Uncertainty, Tempering, and Data Augmentation in Bayesian Classification

The Appendix is structured as follows:

- In Appendix A, we discuss broader impacts and limitations of our work.

- In Appendix B, we provide more context for our key observations in the main text.

- In Appendix C, we give more description of stochastic gradient langevin dynamics (SGLD) which we use for our inference procedure in this work.

- In Appendix D, we describe the gaussian approximation to the noisy dirichlet model of Milios et al. [43].

- In Appendix E, we describe another synthetic experiment relating data augmentation, inference, and posterior tempering.

- In Appendix F, we describe how to sharpen the softmax likelihood with logit smoothing, which is our smoothed softmax baseline.

- In Appendix G, we further connect cold and tempered likelihood posteriors.

- In Appendix H, we visualize the noisy dirichlet likelihood for observed and unobserved classes.

- In Appendix I, we extend the interplay of SGHMC with data augmentation (from Section 5) to variational inference.

- In Appendix J, we give preliminary description to a proper likelihood accounting for data augmentation.

- In Appendix K, we describe further experimental results on synthetic problems and the interplay of cold and tempered posteriors.

- In Appendix L, we describe experimental details for all of our experiments.

## A  Limitations and Broader Impacts

**Broader Impacts**

Our work is primarily theoretical and not applied in nature so we do not see any specific negative or societal effects from our work. However, Bayesian deep learning has been applied to both medical diagnoses and autonomous driving [33], while our work is intended to make these types of modelling more robust and reliable by making the inferred probabilities more accurate.

**Limitations**

We see the following limitations for our work:

- As the cold posterior effect has primarily been demonstrated on image datasets, our work almost exclusively focuses on residual architectures on noiseless image datasets.

- Our work primarily focuses on SGHMC (although we show similar results for SVI) under data augmentation due to the computational expense and difficulties of running HMC with large models under data augmentation [31].

- Our work solely focuses on understanding the interactions between data augmentation and posterior tempering, but does not provide guarantees of robustness to label noise or fixes to other modelling issues.

# B Our Observations in the Context of Prior Work

**Noci et al. [48]: Data augmentation is sufficient but not necessary for CPE.** Noci et al. [48] confirm the findings in Izmailov et al. [31] and Fortuin et al. [14] that data augmentation plays a significant role in the cold posterior effect. In particular, Izmailov et al. [31] show that removing data augmentation from the models in Wenzel et al. [61] alleviates the cold posterior effect in all of their examples. However, Noci et al. [48] also show that it is sometimes possible to achieve CPE without data augmentation, for example by subsampling the dataset. Similarly, Wilson and Izmailov [62] note that many types of misspecification could lead to a cold posterior effect.

In our work, we provide a more nuanced perspective on the cold posterior effect and data augmentation: data augmentation leads to a poor representation of aleatoric uncertainty, which can be addressed by tempering the posterior. We note that BNNs can misrepresent aleatoric uncertainty even without data augmentation, and can have a reasonable representation of aleatoric uncertainty in the presence of data augmentation. However, in the vast majority of practical scenarios where the cold posterior effect has been demonstrated, it appears due to the use of data augmentation. Thus data augmentation is *largely* responsible for the cold posterior effect, and we have shown here that this is precisely because augmentation changes our representation of aleatoric uncertainty to be underconfident, and tempering can correct for this underconfidence, more honestly representing our beliefs.

Finally, Izmailov et al. [30] show that BNNs have issues under covariate shift, due to the posterior not contracting sufficiently along some directions in the parameter space. The same issue occurs when BNNs are applied to extremely small datasets, which may affect the results on data subsampling presented in Noci et al. [48].

**Nabarro et al. [45]: the lack of a CPE without DA is an artifact of using the wrong model (i.e. without DA).** Similar to our discussion in Section 5, Nabarro et al. [45] point out that tempering does not always provide a principled approach to data augmentation. They instead devise a new observation model, where the model outputs are averaged over all the augmented datapoints. With this observation model, they still observe the cold posterior effect. These results are interesting in that they show that a valid likelihood is not sufficient to remove the cold posterior effect. However, these results do not rule out the possibility that the proposed likelihood is a poor description of the data generation process: indeed, in reality the labels for the training image are not produced by considering all the possible augmentations of these images.

Generally, our results suggest that the cold posterior effect is caused by a poor representation of the aleatoric uncertainty in the BNNs. It can happen with or without data augmentation, with valid or invalid likelihoods. A valid likelihood is not necessarily a well-specified likelihood. However, we believe that the interpretation that the lack of CPE in models without data augmentation is an artifact of misspecification is highly questionable: when our beliefs about the aleatoric uncertainty in the data are correctly captured by the model, we should not need posterior tempering to achieve strong results; it is the need for tempering which is an indication of misspecification.

**Noci et al. [48]: Cold posteriors are unlikely to arise from a single simple cause.** Noci et al. [48] show examples where the cold posterior effect arises in isolation from data augmentation, data curation, or prior misspecification. Consequently, they argue that no one cause can fully explain the cold posterior effect. While we agree with this general argument, which is also aligned with the discussion in Wilson and Izmailov [62], we note that all of the considered causes are directly related to aleatoric uncertainty. Indeed, in this work we have shown that data augmentation significantly affects the level of aleatoric uncertainty assumed by the model. The data curation is also directly connected to aleatoric uncertainty, as curated datasets are expected to have low label noise. Finally, in BNNs the prior over the parameters specifies the assumptions about aleatoric uncertainty, and tempering can be used to correct for the effect of the prior. We believe that our results in this paper provide a compelling explanation for the observations in Noci et al. [48].

**Summary.** Overall, properly representing aleatoric uncertainty is a challenging but fundamentally important consideration in Bayesian classification. We have shown that posterior tempering provides a mechanism to more honestly represent our beliefs about aleatoric uncertainty, especially in the presence of data augmentation. In general, as in Wilson and Izmailov [62], we should not be alarmed if $T = 1$ is not optimal in sophisticated models on complex real-world datasets. Moreover, we have shown how other mechanisms to represent aleatoric uncertainty, such as the noisy Dirichlet model,

do not suffer from a cold posterior effect in the presence of data augmentation. Indeed, while an interesting phenomenon, cold posteriors should not be conflated with the success or failure of Bayesian deep learning. In general, approximate inference in Bayesian neural networks has been making great strides forward, often providing better performance at a comparable computational cost to standard methods. However, there *are* practical challenges to the adoption of Bayesian deep learning. For example, Izmailov et al. [30] shows that Bayesian neural networks can profoundly degrade in performance under a wide range of relatively minor distribution shifts — behaviour which could affect applicability on virtually any real-world problem, since train and test rarely come from exactly the same distribution. While their *EmpCov* prior provides a partial remedy, there is still much work to be done. For example one could develop priors that protect against covariate shift by accounting for linear dependencies in the internal representations of the network.

## C  Stochastic Gradient Langevin Dynamics (SGLD).

For large-scale neural networks, exact posterior inference remains intractable. To approximate samples from the posterior $p(w \mid \mathcal{D})$ over parameters of a neural network, we simulate the time-discretized Langevin stochastic differential equation (SDE) [55, 60]

$$
\begin{aligned}
\Delta w &= \mathbf{M}^{-1}\mathbf{m}\epsilon, \\
\Delta \mathbf{m} &= -\nabla_w \widetilde{U}(w)\epsilon - \gamma\mathbf{m}\epsilon + \sqrt{2\gamma T}\eta, \ \text{where } \eta \sim \mathcal{N}\left(\mathbf{0}, \mathbf{M}\right),
\end{aligned}
\tag{12}
$$

where $\mathbf{m}$ are the auxiliary momentum variables, $\mathbf{M}$ is the mass matrix which acts as a preconditioner (often identity), $\gamma$ is the friction parameter, $T$ is the temperature, $\Delta t = \epsilon$ is the time discretization (step size), and $\nabla_w \widetilde{U}(w)$ is an unbiased estimator of the gradient $\nabla_w U(w)$ using only a subset of the dataset $\mathcal{D}$ for computational efficiency. For $\epsilon \to 0$ in the limit of time $t \to \infty$, simulating Eq. (12) produces a trajectory distributed according to the stationary distribution $\exp\{-U(w)/T\}$, which is exactly the posterior

$$
p_{\text{cold}}(w \mid \mathcal{D}) \propto \exp\left\{ -\frac{1}{T} \underbrace{\left(-\log p(\mathcal{D} \mid w) - \log p(w)\right)}_{U(w)} \right\},
\tag{13}
$$

we desire in Section 3. When $\gamma = 0$, Eq. (12) describes the Stochastic Gradient Langevin Dynamics (SGLD) [60], and otherwise it describes the Stochastic Gradient Hamiltonian Monte Carlo (SGHMC) where $1 - \gamma$ represents the momentum [8]. Furthermore, we can sample from Eq. (3) by setting $T = 1$ in Eq. (12) and raising only the likelihood to a power $T$. In addition, it is often beneficial to use a cyclical time-stepping schedule for $\epsilon$ in Eq. (12), as proposed by Zhang et al. [65] for cyclical-SGLD (cSGLD) when $\gamma = 0$, and cyclical-SGHMC (cSGHMC) when $\gamma > 0$.

## D  Gaussian Approximation to the noisy Dirichlet Model.

Milios et al. [43] considered the distribution in Eq. (9) in the context of Gaussian process classification, but with a different goal: they aimed to create a regression likelihood which would approximate the softmax likelihood. They further approximated the Dirichlet distribution $\text{Dir.}(f(x))$ over the class probabilities $f(x)$ with a product of independent Gaussian distributions over the logits $z(x)$:

$$
p_{NDG}(w \mid D) \propto p(w) \prod_{x,y \in \mathcal{D}} \prod_{c=1}^{C} \mathcal{N}(z_c(x) \mid \mu_c, \sigma_c^2), \ \text{with}
\tag{14}
$$

$$
\alpha_c = 1 + \alpha_\epsilon \cdot I[c = y], \ \ \sigma_c^2 = \log(1/\alpha_c + 1), \ \ \mu_c = \log(\alpha_c) - \frac{\sigma_c^2}{2},
$$

where $I[c = y]$ is the indicator function equal to 1 for $c = y$ and 0 for $c \neq y$. Here *NDG* stands for *noisy Dirichlet Gaussian* approximation. While theoretically we do not need to use the Gaussian approximation in Eq. (14) for Bayesian neural networks and can directly use Eq. (9), we found that in practice the approximation is much more stable numerically. Indeed, the approximation in Eq. (14) amounts to solving a regression problem in the space of logits $z(x)$ with a Gaussian observation model. In the experiments in this paper we will use the $p_{NDG}$ model.

**Visual comparison.** In Figure 6 (left), we show the CDF of the posterior distribution over the confidence $f_y$ for the correct class $y$ with the standard softmax likelihood, tempered likelihood and

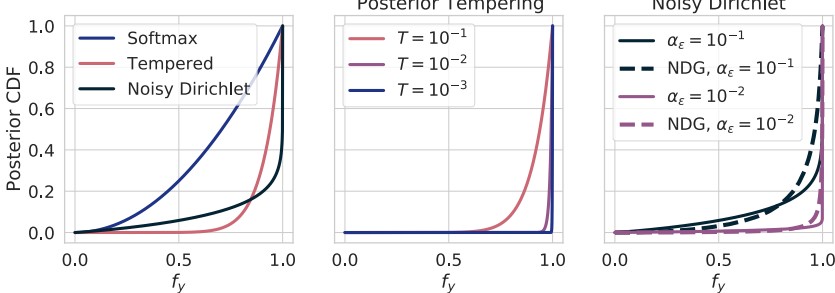

Figure 6: **Comparison of Bayesian classification models.** The probability CDF of the posterior over the confidence $f_y$ for the correct class $y$ with the standard cross entropy likelihood, tempered likelihood and the noisy Dirichlet model in a binary classification problem. **Left**: Tempering and noisy Dirichlet models both allow to concentrate the posterior on solutions that are confident in the correct label. **Middle**: Lower likelihood temperatures lead to higher concentration on confident solutions. **Right**: In the noisy Dirichlet model, lower values of the noise parameter $\alpha_\epsilon$ also lead to higher confidence.

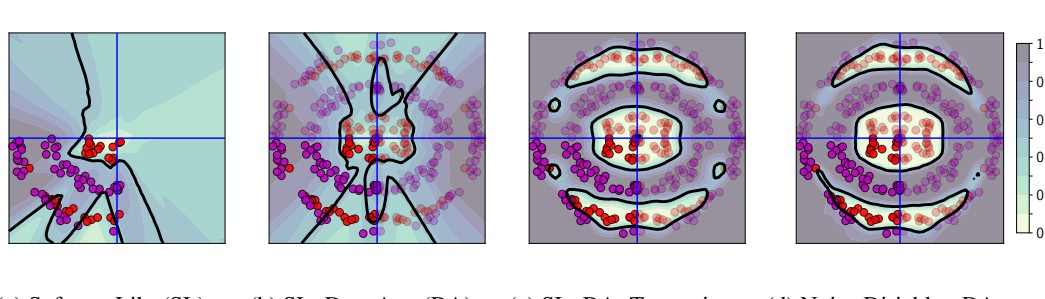

(a) Softmax Lik. (SL)   (b) SL+Data Aug.(DA)   (c) SL+DA+Tempering   (d) Noisy Dirichlet+DA

Figure 7: **Insufficient Sharpness of Softmax Likelihood.** Decision boundary of BNN classifiers on a synthetic problem. In each panel, the decision boundary is shown with a black line, datapoints are shown with magenta (class 0) and red (class 1) circles. We consider data augmentation – random flip about the axes shown with blue lines; the augmented datapoints are shown semi-transparently. We use HMC to approximate the posterior in each of the panels. **(a)** A standard softmax likelihood provides a reasonable but imperfect fit of the training data. **(b)** Adding data augmentation leads to a more diffuse likelihood which is underconfident, and therefore an even worse fit. **(c)** Tempering with $T = 0.1$ sharpens the softmax likelihood leading to a perfect fit on the train data. **(d)** A noisy Dirichlet model, provides a similar fit without the need for tempering.

the noisy Dirichlet model for a binary classification problem. Here, we assume that the prior over $\{f(x)\}_{x \in \mathcal{D}}$ is uniform and independent across $x$ for the purpose of visualization. Both tempering and the noisy Dirichlet model indeed push the predictions $f_y$ to be much more confident in the correct class $y$ compared to the standard softmax likelihood. In Figure 6 (middle) we show the effect of the value of temperature: lower temperatures correspond to more confident predictions. Similarly, in Figure 6 (right) lower values of $\alpha_\epsilon$ lead to more confident predictions in the noisy Dirichlet model. The Gaussian approximation NDG matches the CDF of the noisy Dirichlet model well, but smooths it slightly, making it more amenable to numerical sampling.

## E   Synthetic Experiment

We generate the data from a modified version of the two spirals dataset [see e.g. 39], where we restrict the data to the $x_1 < 0, x_2 < 0$ quadrant. The exact code used to generate the data is available in the Appendix L.1.

In Figure 7(a) we visualize the data and the decision boundary of a Bayesian neural network. For the Bayesian neural network, we use an iid Gaussian prior $\mathcal{N}(0, 0.3^2)$ over the parameters of the

model; we use full batch Hamiltonian Monte Carlo (HMC) to sample from the posterior, following Izmailov et al. [31]. With the chosen prior, the model is not able to fit the data perfectly, but provides a reasonable fit of the training data.

In Figure 7(b) we consider the effect of data augmentation. Specifically, we apply random flips about the $x_1$ and $x_2$ axes. In the figure, the augmented datapoints are shown semi-transparently. To evaluate the network under data augmentation, we run HMC to sample from the posterior distribution in Eq. (11). The Bayesian neural network provides a lower quality fit to the data compared to the fit without data augmentation shown in Figure 7(a). Indeed, according to our analysis in Section 5, the observation model is softened by the data augmentation, leading the model to fit the training data poorly.

According to Section 4, we can sharpen the model leading to a much better fit on the train data by using the tempered likelihood as in Figure 7(c) or the noisy Dirichlet observation model as in Figure 7(d). With both of these approaches we achieve a near-perfect fit on the training data, which is desirable if we assume low aleatoric unceratinty. For further details on this experiment, please see Appendix L.1.

To intuitively grasp the impact of aleatoric uncertainty on inference, we first develop a toy illustration. Consider the *two spirals* binary classification problem [28] where, for training we generate 50 samples with both $x_1, x_2 < 0$. Figure 7 visualizes the posterior predictive density for various settings.

By naively learning from the training data using the softmax likelihood, we find that the predictive density is poorly calibrated to the effect that it does not learn anything meaningful except in the region containing the training data, as visualized in Figure 7a. But, by virtue of our a priori knowledge about the mirror symmetry across both axes, we can create augmented data, i.e. for every $(x, y)$ pair in the training data, we create augmentations $(\pm x, \pm y)$ for use in training. With this modification, we find that the predictive density surface in Figure 7b appears to improve but remains very diffuse, implying underconfidence.

But in fact, we can fix the predictive density surface (and hence the performance) by accounting for aleatoric uncertainty. In Figure 7c we see that tempering allows us to sharpen the softmax likelihood such that the final predictive density does not remain diffuse. We are implicitly correcting for aleatoric uncertainty which was artificially inflated due to data augmentations. Alternatively, by using a noisy Dirichlet likelihood which allows explicit control over the amount of aleatoric uncertainty in our data observation model, we are able to achieve a similar effect as shown in Figure 7d.

This toy illustration is representative of how modern machine learning models for classification are trained in practice, and provide a first direct understanding of the interaction of aleatoric uncertainty with tempering and data augmentation.

## F    Sharpening with the Smoothed Softmax Likelihood

A natural approach to consider sharpening the softmax likelihood is by smoothing the logits. The *tempered softmax likelihood* for a given class observation $y = c$ conditional on input $x$ is given by,

$$\frac{1}{T} \log p(y = c \mid x) = \log \frac{\exp\{f_c(x; w)/T\}}{(\sum_{j=1}^{C} \exp\{f_j(x; w)\})^{1/T}} \tag{15}$$

where $f_j(x; w) \in \mathbb{R}$ represents the logit for the $j^{\text{th}}$ class of a total of $C$ classes. A properly normalized distribution, however, leads to what we term the *smoothed softmax likelihood*,

$$\log p_T(y = c \mid x) = \log \frac{\exp\{f_c(x; w)/T\}}{\sum_{j=1}^{C} \exp\{f_j(x; w)/T\}}. \tag{16}$$

Versions of this likelihood have appeared in prior literature, especially when considering distillation [24] and temperature (e.g. Platt) scaling [52, 19]. More recently, Zeno et al. [64] have pointed out a connection between Eq. (16) and the cold posteriors Eq. (3) studied by Wenzel et al. [61].

**Insufficiency of smoothed softmax.**    Consider a toy coinflip example — we are interested in inferring the probability of heads $w \in [0, 1]$ using a single observation of a heads $\mathcal{D} = \{H\}$.

The posterior over $w$, under the assumption of a uniform prior $p(w) = \mathrm{U}[0,1]$ is given by $p(w \mid \mathcal{D}) \propto p(H \mid w) = \theta$. The smoothed softmax likelihood corresponds to rescaling the logit $\ell(w) = \log\{w/(1-w)\}$ as $p_T(H \mid w) = \sigma(\ell(w)/T)$, where $\sigma(z) = 1/(1 + \exp\{-z\})$ is the sigmoid function [6].

Using the tempered softmax likelihood by raising to a power $1/T$ where $T < 1$, we effectively sharpen the density function and force the posterior to concentrate more sharply around 1 owing to the single observation. On the other hand, using a smoothed softmax likelihood ends up spreading the posterior mass very differently, to the effect that it undermines the evidence from observation. This behavior is visualized in Figure 8. In fact, no matter what temperature we use, the smoothed softmax likelihood cannot be made arbitrarily sharp.

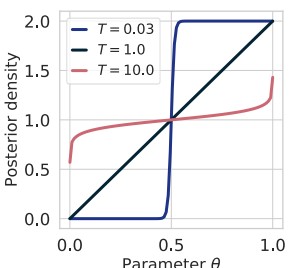

In Figure 10, we empirically verify with ResNet-18 on CIFAR-10 that no matter the value of temperature $T$, a smoothed softmax likelihood does not outperform a tempered likelihood. Moreover, it does not mitigate the cold posterior effect. Therefore, we can return to our discussion of cold and tempered posteriors.

**Smoothed softmax and reparameterization.** Intuitively, for Bayesian neural networks, the smoothed softmax likelihood corresponds to a simple reparameterization of the standard softmax likelihood BNN, where the last layer parameters are all divided by a scalar constant $T$. This model is equivalent to simply rescaling the prior over the last layer parameters. In particular, for BNNs with zero-mean iid Gaussian priors, this reparameterization is equivalent to dividing the standard deviation of the prior over the last layer parameters by $T$. Such rescaling has a very limited effect on the properties of the model, especially given the homogeneity of the ReLU activations and the scale invariance due to normalization layers, which may provide further intuition for why the smoothed softmax likelihood is insufficient for modeling aleatoric uncertainty in Bayesian classification.

Figure 8: A smoothed likelihood Eq. (16) is insufficient to achieve arbitrary sharpness (Appendix F). An arbitrarily sharp likelihood would concentrate around a single value of $\theta$ rather than being effectively uniform over ranges.

## G  Connecting Cold Posteriors and Tempered Likelihood Posteriors

In this appendix, we explicitly spell out the connections between cold, tempered, and Bayesian posteriors, focusing first on the shape of the posterior with Bayesian linear models before moving into describing how the prior changes with a cold posterior.

### G.1  Connections in Bayesian Linear Models

We begin by taking a look at the connection between the Bayesian, cold, and tempered posteriors under linear models. For a dataset $\mathcal{D} = \{X, y\}$ consisting of $d$-dimensional inputs $X = [x_1; x_2; \ldots; x_N] \in \mathbb{R}^{N \times d}$, corresponding outputs $y \in \mathbb{R}^N$, a known observation noise variance $\sigma^2$, and prior over parameters $w \sim \mathcal{N}(0, \alpha \mathbf{I})$, the posterior over parameters is given by,

$$p(w \mid \mathcal{D}) = \mathcal{N}(\left(X^\top X + \alpha^{-1} \sigma^2 \mathbf{I}\right)^{-1} X^\top y, \left(\sigma^{-2} X^\top X + \alpha^{-1} \mathbf{I}\right)^{-1}). \tag{17}$$

Following Grünwald and Van Ommen [18], who study $T > 1$ in the context of model misspecfication, the tempered posterior for an explicit temperature $T$ is,

$$p_{\text{temp}}(w \mid \mathcal{D}) = \mathcal{N}(\left(\tfrac{1}{T} X^\top X + \alpha^{-1} T \sigma^2 \mathbf{I}\right)^{-1} X^\top y, \left(\tfrac{1}{T} \sigma^{-2} X^\top X + \alpha^{-1} \mathbf{I}\right)^{-1}). \tag{18}$$

Further, for a *cold posterior*, where both the likelihood and the prior are raised to a temperature, we can follow Aitchison [2] and rewrite the prior over $w$ as $\mathcal{N}(0, \alpha T \mathbf{I})$, giving the posterior as,

$$p_{\text{cold}}(w \mid \mathcal{D}) = \mathcal{N}(\left(X^\top X + \alpha^{-1} \sigma^2 \mathbf{I}\right)^{-1} X^\top y, T\left(\sigma^{-2} X^\top X + \alpha^{-1} \mathbf{I}\right)^{-1}). \tag{19}$$

Thus, "cold" posteriors in this setting reproduce the same posterior mean as the standard Bayesian posterior, only varying the value of the posterior covariance matrix. We visually demonstrate this effect for the posterior confidence ellipses over $\theta$ in Figure 9 left.

---

[6]Note here that $\sigma(\ell(\theta)) = \theta$

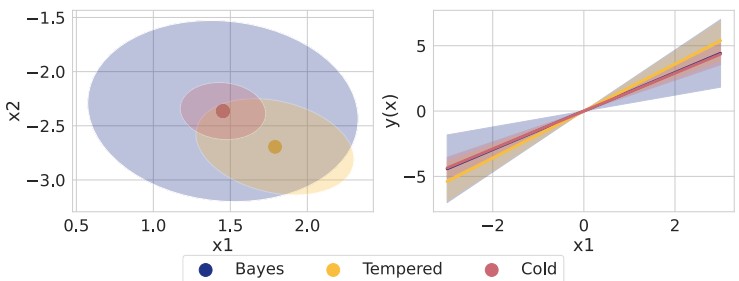

Figure 9: **Left:** Posterior confidence ellipses for Bayesian linear regression in both Bayesian, tempered, and cold frameworks. **Right:** Posterior functions for all three approaches. The cold posterior simply rescales the prior, producing more confident fits.

For $T < 1$, we tend to increase the concentration of the posterior around the posterior mean. In contrast, for $T > 1$, we tend to decrease the concentration of the posterior around the posterior mean. Propagating the posterior distribution on $w$ to the resulting distribution over the function $y_\star = w^\top x_\star$ for novel inputs $x_\star$, the cold posterior over $y_\star$ has the same mean as the Bayesian posterior, while only the posterior covariance is rescaled by a factor of $T$. We also demonstrate this effect in Figure 9 right, showing that the slope of the cold model is the same as the slope of the fully Bayesian model, only their posterior variances are different. However, the slope of the tempered model is different, owing to an effectively increased noise term.

A similar finding is produced by Adlam et al. [1] in the context of Gaussian process regression, where similarly cold posteriors simply rescale the posterior variance of the GP posterior, while tempering the posterior tends to increase the noise level.

## G.2 Cold Posteriors and Sharp Likelihoods

While constructing a cold posterior Eq. (2) involves tempering both the likelihood and the prior, the cold posterior effect still remains even with a tempered likelihood posterior Eq. (3) alone. Many classes of prior distributions, specifically any prior that is bounded almost everywhere, continue to be proper prior distributions when raised to a power $1/T$. We formalize this statement in Theorem 1.

**Theorem 1.** *For proper prior distributions, $p(w)$, that have bounded density functions, e.g. $p(w) \le M$ almost everywhere, then for $T \le 1$, $p(w)^{1/T}/\mathcal{I}$ is a proper prior distribution, where $\mathcal{I} = \int p(\theta)^{1/T} d\theta < \infty$.*

*Proof.* As $p(\theta)$ is bounded above by $M$, then $p(\theta)/M \le 1$ for all $x$. As $1/T \ge 1$, then $(p(\theta)/M)^{1/T} \le p(\theta)/M \le 1$. By construction $p(\theta)/M$ is integrable as $p(w)$ is integrable, so then $(p(w)/M)^{1/T}$ must also be integrable implying that $p(w)^{1/T}$ is integrable. To make $p(w)^{1/T}$ a distribution, we only need to construct a normalization constant $\mathcal{I}$. ∎

Theorem 1 shows that the cold posterior effect in the Bayesian deep learning literature can be equivalently thought of in terms of a *tempered likelihood* posterior alone. The only difference is that the actual priors being used are different from the stated priors in that they have tighter variances. See Appendix G for further examples and discussion.

The above result unifies several tempered likelihood findings [65, 21] to those with the cold posterior studies [61, 14]. Using a ResNet-18 on CIFAR-10, we empirically demonstrate such an equivalence in Figure 10. A rescaled prior and step size $\epsilon$ can match the performance of a cold posterior. To match the gradient scaling, we must downscale the step size by the square root of the

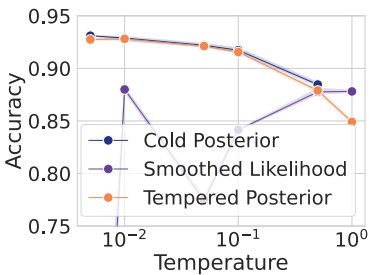

Figure 10: ResNet-18 on CIFAR-10. Cold posteriors are matched by tempered posteriors, but not by the smoothed likelihood.

implied temperature, while we match the prior variance by setting it to $\epsilon T$. We give specific examples of these priors in the next section.

Thus, to understand the cold posterior effect, we may focus on the tempered likelihood term in Eq. (3). Tempering the likelihood with $T < 1$ is sharpening the density function to facilitate a more accurate observation model per our beliefs. It is then only natural to use a likelihood which allows direct control of the sharpness, which we introduce next. We also note that the likelihood and the prior tend to interact here. For a fixed prior, a sharper likelihood will tend to produce a sharper posterior, while decreasing the variance on the prior will additionally tend to produce a sharper (more confident) posterior. This is exactly the effect produced by likelihood tempering with $T < 1$; tempering the likelihood produces a sharper likelihood which therefore produces a sharper posterior.

**More Cold Priors**

Theorem 1 covers many common prior choices in the Bayesian deep learning literature. For *Gaussian and Laplace priors*, the scaling term (variance in the case of the Gaussian prior and scale in the case of the Laplace prior) changes by a factor of $\epsilon T$ where $\epsilon$ is the original scale. It also holds for correlated Gaussian priors such as those studied by Fortuin et al. [14], Izmailov et al. [30]. This effect was previously noted for Gaussian priors by Aitchison [2].

For *Student's t priors*, we show below that the degrees of freedom increases and the scale term changes producing a distribution with smaller variance than the original distribution. Similarly for *Cauchy priors*, the entire form of the distribution changes and in such a way that it now has finite mean and variance, which it did not before.

The log pdf for a non-central student T distribution[7] is

$$\log p(x) = \log \Gamma(v + 1/2) - \log \Gamma(v/2) - \log \sqrt{v\pi}\sigma - \frac{v+1}{2} \log \left( 1 + \frac{x^2}{v\sigma^2} \right)$$

Rescaling by a temperature $T$ gives

$$\frac{1}{T} \log p(x) = \frac{1}{T} \left( \log \Gamma(v + 1/2) - \log \Gamma(v/2) - \log \sqrt{v\pi}\sigma \right) - \frac{v+1}{2T} \log \left( 1 + \frac{x^2}{v\sigma^2} \right)$$

and letting $\tilde{v} = \frac{v+1}{T} - 1$ produces

$$\frac{1}{T} \log p(x) = \text{const.} - \frac{\tilde{v}+1}{2} \log \left( 1 + \frac{\tilde{v}}{\tilde{v}} \frac{x^2}{v\sigma^2} \right) = \text{const.} - \frac{\tilde{v}+1}{2} \log \left( 1 + \frac{x^2}{\tilde{v}\tilde{\sigma}^2} \right)$$

with $\tilde{\sigma}^2 = v\sigma^2/\tilde{v} = \frac{v\sigma^2 T}{v+1-T}$.

For $v > 2$, the original variance was $\sigma^2 \frac{v}{v-2}$ and now it is $\sigma^2 \frac{T}{v+1-3T}$, which is less whenever $v > 2$ and so the variance exists. Note additionally that $\tilde{v} > v$ as well.

For a Cauchy distribution (or equivalently, follow the argument for the non-central $t$ with $\eta = 1$), the tempered prior follows the form

$$p_T(x) \propto (1 + x^2)^{-1/T},$$

which produces a finite integral and therefore a proper prior for all $T \geq 1$. This class of distributions has finite means for any $T < 1$ and finite variances for $T < 2/3$, whereas the Cauchy distributions do not have finite means or variances. This change may have the effect of making posterior means exist for cold posteriors when they may not have existed before, as can be the case for logistic regression with Cauchy priors [16].

### G.3 Discussion with the Cold Posterior Effect

These types of findings tend to suggest that "cold" priors have smaller variances (or spreads, if the variances do not exist) than their warm counterparts. This is most clearly the case for Gaussian and Laplace prior distributions where the prior variance is reduced from $\epsilon$ to $\epsilon T$. One natural question is if this finding is an explanation for the "cold" posterior effect, an analogue of prior-misspecification,

---

[7] https://en.wikipedia.org/wiki/Noncentral_t-distribution

as discussed by Wenzel et al. [61]. However, Wilson and Izmailov [62] found that standard Gaussian $\mathcal{N}(0, 1)$ priors tend to be reasonably well-calibrated and that varying priors tends to have minimal impact on down-stream accuracy and calibration. Izmailov et al. [31] verified this evidence with high-quality HMC samples, but without data augmentation.

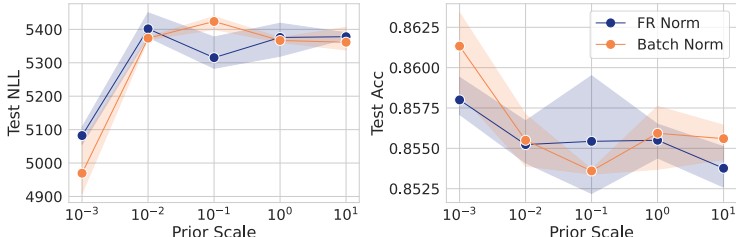

Figure 11: Priors of various scales for BNNs with temperature 1. No prior with the same step size is able to match the performance of the lower temperature.

In Figure 11, we show that no choice of prior alone with temperature 1 is able to reach the same accuracy as a tempered (or cold) baseline model. Thus, it is not merely (Gaussian) prior misspecification that causes cold posteriors to exist, and we need to look beyond this potential explanation.

# H    Visualizations of Noisy Dirichlet Likelihoods

In Figure 12, we show various visualization of the likelihood for the observed and unobserved classes for the noisy Dirichlet likelihood.

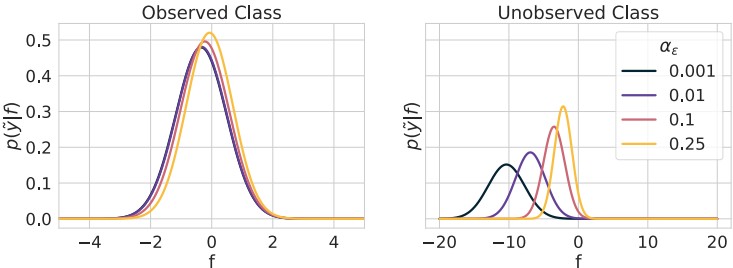

Figure 12: Dirichlet densities for various values of $\alpha_\epsilon$ for both the observed class (**left**) and any un-observed classes (**right**). Higher values of $\alpha_\epsilon$ tend to produce sharper likelihoods as the distributions corresponding to these values have lower variances. Here $f$ is the equivalent to the logit, while $\tilde{y}$ is the rescaled observation.

# I    Targeting Distributions for Variational Inference

A naive attempt to incorporate data augmentation would be to simply stack all $K$ copies of the data into the posterior and then to infer it; however, this approach would contract the posterior by too much as we would be considering $NK$ data points rather than $N$ data points. A second approach would be then to down-weight the augmentations so that we effectively see only $N$ data points by raising the likelihood of each (augmented) data point to a power $1/K$, which produces exactly Eq. (11).

Variational inference with stochastic gradient descent using the evidence lower bound (ELBO) produces a similar gradient to stochastic gradient descent as

$$\nabla \text{ELBO}(\phi) = \frac{N}{m} \sum_{(x_i, y_i) \in \mathcal{D}_m} \mathbb{E}_{q_\phi(w)} \nabla_w \log p(y_i \mid t_j(x_i)) + \nabla_\phi \text{KL}(q_\phi(w) \| p(w)), \qquad (20)$$

switching the parameters of the variational distribution to $\phi$ and again with transformations $t_j$ sampled uniformly from $\mathcal{T}$. The same two sources of randomness, sub-sampling and augmentation,

apply in this situation and computing the expected value of Eq. (20) produces a similar target as Eq. (10):

$$\mathbb{E}[\nabla \text{ELBO}(\phi)] = \sum_{i=1}^{N} \sum_{j=1}^{K} [\nabla_w \log p(y_i \mid t_j(x_i))^{1/K}] + \nabla_\phi \text{KL}(q_\phi(w) || p(w)), \tag{21}$$

implying that the full optimization problem VI is targeting uses a tempered likelihood in that term, e.g. $p(y_i \mid t_j(x_i))^{1/K}$. We note that variational inference for BNNs often does end up requiring tempering by downweighting the KL term in the ELBO, see Section 2.3 of Wenzel et al. [61] for several examples. While we do not consider Laplace approximations under data augmentation, they additionally tend to require some amount of likelihood down-weighting for high performance (see e.g. Immer et al. [29] for example), suggesting that they also tend to target a similar likelihood distribution.

## J A Proper Likelihood for Data Augmentation

Treating data augmentations as independent samples from the true underlying data generating process is problematic, since the augmented samples are highly correlated with the originally observed samples. In Section 6.1, we see that stacking the augmented samples in to a single unified dataset is problematic in the sense that observation noise artificially inflated. For well-calibrated predictions, this is a mischaracterization of the data generating process.

To arrive at a proper likelihood accounting for data augmentations $t_j \in \mathcal{T}$, consider a likelihood which extends the usual observation likelihood with a consistency term,

$$p_{\text{aug}}(\mathcal{D} \mid w) = \prod_{i=1}^{N} p(y_i \mid x_i) \cdot \prod_{t_j \in \mathcal{T}} p\left(f(t_j(x_i); w)) \mid f(x_i; w)\right). \tag{22}$$

The augmentation likelihood $p_{\text{aug}}$ properly accounts for aleatoric uncertainty when the augmentation is trivial, and not lead to underconfident predictions when the augmentations are uncorrelated. For a minibatch sample from the full dataset $\mathcal{D}_m \subset \mathcal{D}$ of size $m$, and a subset of $k$ augmentations $\mathcal{T}_k \subset \mathcal{T}$, the valid unbiased stochastic gradient estimator for use with SGLD is given by,

$$\nabla \widetilde{U}_{\text{aug}}(w) = \frac{N}{m} \sum_{(x_i, y_i) \in \mathcal{D}_m} \nabla_w \log p(y_i \mid t_k(x_i)) + \nabla_w \log p(w)$$
$$+ \cdot \frac{N}{m} \sum_{x_i \in \mathcal{D}_m} \frac{K}{k} \sum_{t_j \in \mathcal{T}_k} \nabla_w \log p\left(f(t_j(x_i); w)) \mid f(x_i; w)\right). \tag{23}$$

Note that to use the augmentation likelihood in Eq. (22), we need to know the number $K$ of possible augmentations that we consider.

## K Further Experimental Results

### K.1 Synthetic Examples

In the top left panel of Figure 13, we demonstrate that softmax scaling in the presence of label noise produces very underconfident predictions.

For GPs, varying the values of $\alpha_\epsilon$ then vary the sharpness of the resulting posterior predictive distribution, as we show in Figure 13 for a two spirals problem with 150 data points and 20 noisy labels. The posterior predictive for $\alpha_\epsilon = 0.001$ is only confident in a very small region of the data, whereas the higher values of $\eta$ are confident in a broader region indicating a sharper likelihood that enables more certainly about the observed data. This is in comparison to a Laplace approximated Bernoulli GP classifier that is less confident about the data than $\alpha_\epsilon = 0.1$ but performs similarly in terms of confidence to the Dirichlet with $\alpha_\epsilon = 0.25$, suggesting that too sharp of distributions can reduce some confidence.

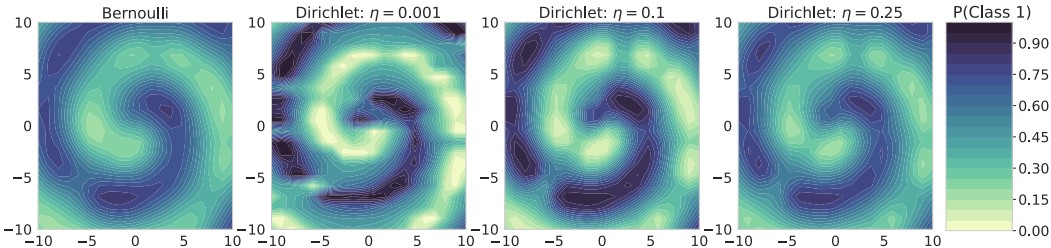

Figure 13: Modeling label noise explicitly using Eq. (14) and using tuned values of $\eta$ enables better modelling of aleatoric uncertainty in the presence of label noise for GP classifiers on a two spirals problem with 150 data points and 20 flipped labels.

## K.2 Label Noise in Image Classification

This section expands on the results presented in Figure 3 by also plotting the sub-optimal values of $T$ and $\alpha_\epsilon$ in Figure 14 to provide the reader a more complete picture of performance on various values of the parameters for both likelihoods considered in this work.

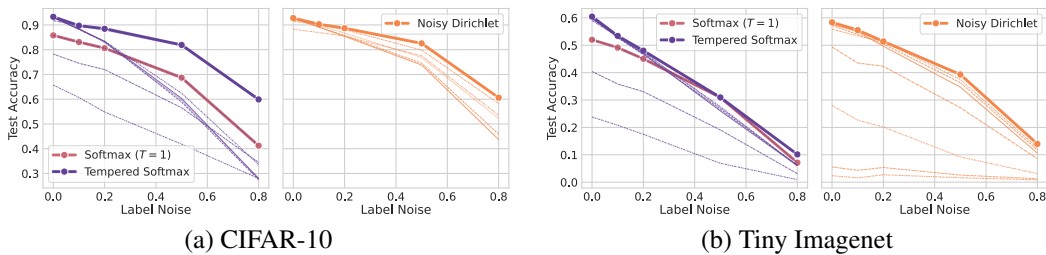

(a) CIFAR-10           (b) Tiny Imagenet

Figure 14: We expand on the results in Figure 3 by plotting performance curves for all temperatures $T$ and noise values $\alpha_\epsilon$, with the optimal curve highlighted. Each thin line in purple represents the performance on temperatures $T \in \{10^{-5}, 10^{-4}, 10^{-3}, 10^{-2}, 0.1, 1., 3., 10.\}$ for the tempered softmax likelihood, and each thin line in orange represents the performance on noise values $\alpha_\epsilon \in \{10^{-6}, 10^{-5}, 10^{-4}, 10^{-3}, 10^{-2}, 0.1\}$ for the noisy Dirichlet likelihood. Numerical data is provided in Tables 1 and 2.

## K.3 Learning Rate Rescaling

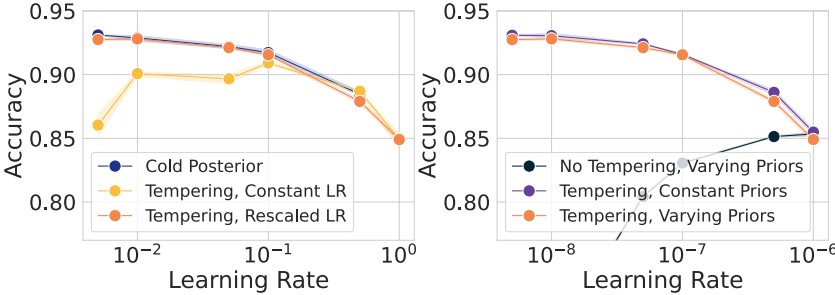

Figure 15: We empirically verify that a tempered likelihood combined with prior rescaling is sufficient to reproduce the cold posterior effect, as discussed in Appendix G.

In Figure 15, we demonstrate that cold posteriors are identical to tempered posteriors when using SGLD with a rescaled learning rate. Again, here, we use ResNet-18 on CIFAR-10. Furthermore, we also need to vary the scale of the Gaussian prior to produce identical results; however the scale of the prior changes the results minimally when using tempering.

| Label Noise | $T$ | Test BMA Accuracy (%) | Test NLL |
|---|---|---|---|
| 0 | 0.00001 | $92.97 \pm 0.19$ | $3577.74 \pm 36.69$ |
| | 0.0001 | $93.13 \pm 0.08$ | $3599.74 \pm 14.91$ |
| | 0.001 | $\mathbf{93.28} \pm 0.08$ | $3476.33 \pm 18.85$ |
| | 0.01 | $92.94 \pm 0.12$ | $3521.37 \pm 32.88$ |
| | 0.1 | $91.77 \pm 0.26$ | $3587.2 \pm 56.83$ |
| | 1 | $85.78 \pm 0.19$ | $5344.59 \pm 31.25$ |
| | 3 | $78.24 \pm 0.13$ | $7891.12 \pm 48.70$ |
| | 10 | $65.69 \pm 0.36$ | $12440.25 \pm 61.04$ |
| 0.1 | 0.00001 | $88.43 \pm 0.08$ | $5872.89 \pm 104.96$ |
| | 0.0001 | $88.15 \pm 0.23$ | $6000.17 \pm 124.29$ |
| | 0.001 | $88.49 \pm 0.35$ | $5885.82 \pm 33.58$ |
| | 0.01 | $88.23 \pm 0.33$ | $5777.52 \pm 97.05$ |
| | 0.1 | $\mathbf{89.71} \pm 0.20$ | $4322.14 \pm 35.02$ |
| | 1 | $83.08 \pm 0.44$ | $6419.33 \pm 69.41$ |
| | 3 | $74.51 \pm 0.15$ | $9088.32 \pm 35.38$ |
| | 10 | $60.70 \pm 1.01$ | $13551.9 \pm 81.70$ |
| 0.2 | 0.00001 | $83.30 \pm 0.37$ | $8346.63 \pm 87.89$ |
| | 0.0001 | $83.21 \pm 0.40$ | $8425.63 \pm 185.47$ |
| | 0.001 | $82.97 \pm 0.57$ | $8345.62 \pm 284.03$ |
| | 0.01 | $83.49 \pm 0.09$ | $7920.99 \pm 103.56$ |
| | 0.1 | $\mathbf{88.45} \pm 0.61$ | $5448.62 \pm 125.64$ |
| | 1 | $80.62 \pm 0.21$ | $7774.44 \pm 27.49$ |
| | 3 | $71.97 \pm 0.46$ | $10406.1 \pm 57.89$ |
| | 10 | $54.85 \pm 1.37$ | $15327.49 \pm 295.44$ |
| 0.5 | 0.00001 | $60.31 \pm 1.10$ | $17986.78 \pm 364.34$ |
| | 0.0001 | $58.87 \pm 0.65$ | $18381.33 \pm 187.42$ |
| | 0.001 | $59.60 \pm 0.33$ | $17816.36 \pm 258.46$ |
| | 0.01 | $62.40 \pm 0.43$ | $15499.53 \pm 168.51$ |
| | 0.1 | $\mathbf{81.91} \pm 0.39$ | $9998.57 \pm 185.99$ |
| | 1 | $68.70 \pm 0.55$ | $13187.17 \pm 73.68$ |
| | 3 | $56.54 \pm 0.40$ | $16126.59 \pm 78.99$ |
| | 10 | $41.85 \pm 0.54$ | $19707.41 \pm 91.98$ |
| 0.8 | 0.00001 | $27.96 \pm 1.22$ | $29862.19 \pm 1241.84$ |
| | 0.0001 | $27.43 \pm 1.36$ | $31027.27 \pm 382.76$ |
| | 0.001 | $28.08 \pm 0.65$ | $29515.91 \pm 637.08$ |
| | 0.01 | $33.56 \pm 1.43$ | $22806.11 \pm 1126.21$ |
| | 0.1 | $\mathbf{59.90} \pm 0.45$ | $17524.06 \pm 81.75$ |
| | 1 | $41.23 \pm 0.36$ | $20839.3 \pm 58.07$ |
| | 3 | $34.44 \pm 1.29$ | $21811.33 \pm 68.24$ |
| | 10 | $27.96 \pm 1.20$ | $23129.04 \pm 21.34$ |

Table 1: Performance of the standard softmax likelihood model on CIFAR-10, used to plot results in Figure 3(a). We show one standard deviation over 3 trials.

## K.4  Corrupted CIFAR-10 Experiments

Finally, in Figure 16, we display the results for both tempered softmax and noisy Dirchlet for ResNet-18s on corrupted CIFAR-10 [22].

## K.5  Data Cardinality and Underfitting

In Figure 17, we show that increasing the cardinality of the dataset improves the training fit, as measured by the average sample NLL (negative log likelihood) over the training set, showing that dataset cardinality alone cannot explain underfitting.

| Label Noise | $\alpha_\epsilon$ | Test BMA Accuracy (%) | Test NLL |
|---|---|---|---|
| 0. | 0.000001 | $92.75 \pm 0.24$ | $8080.56 \pm 131.65$ |
|  | 0.00001 | $92.72 \pm 0.10$ | $6874.20 \pm 36.90$ |
|  | 0.0001 | $\mathbf{92.78} \pm 0.09$ | $5586.79 \pm 86.25$ |
|  | 0.001 | $92.15 \pm 0.07$ | $4491.19 \pm 117.73$ |
|  | 0.01 | $91.50 \pm 0.07$ | $3564.69 \pm 50.87$ |
|  | 0.1 | $88.26 \pm 0.07$ | $6384.59 \pm 15.93$ |
| 0.1 | 0.000001 | $89.11 \pm 0.29$ | $11507.79 \pm 345.42$ |
|  | 0.00001 | $89.11 \pm 0.24$ | $9638.78 \pm 324.75$ |
|  | 0.0001 | $89.70 \pm 0.06$ | $7399.10 \pm 64.58$ |
|  | 0.001 | $\mathbf{90.28} \pm 0.37$ | $5195.21 \pm 48.68$ |
|  | 0.01 | $90.25 \pm 0.39$ | $3955.32 \pm 77.80$ |
|  | 0.1 | $86.93 \pm 0.05$ | $7194.98 \pm 24.86$ |
| 0.2 | 0.000001 | $85.36 \pm 0.33$ | $14221.11 \pm 161.47$ |
|  | 0.00001 | $85.72 \pm 0.40$ | $11884.17 \pm 241.67$ |
|  | 0.0001 | $86.71 \pm 0.24$ | $8865.18 \pm 187.91$ |
|  | 0.001 | $88.09 \pm 0.18$ | $6006.20 \pm 83.82$ |
|  | 0.01 | $\mathbf{88.68} \pm 0.22$ | $4471.03 \pm 62.10$ |
|  | 0.1 | $85.48 \pm 0.01$ | $8144.23 \pm 11.36$ |
| 0.5 | 0.000001 | $73.80 \pm 1.12$ | $19231.71 \pm 760.60$ |
|  | 0.00001 | $74.54 \pm 0.99$ | $16510.63 \pm 755.69$ |
|  | 0.0001 | $76.85 \pm 0.33$ | $12659.80 \pm 540.86$ |
|  | 0.001 | $79.86 \pm 0.29$ | $8708.74 \pm 136.45$ |
|  | 0.01 | $\mathbf{82.56} \pm 0.19$ | $6995.20 \pm 54.81$ |
|  | 0.1 | $77.55 \pm 0.54$ | $12331.45 \pm 26.44$ |
| 0.8 | 0.000001 | $43.71 \pm 2.93$ | $31334.21 \pm 2914.70$ |
|  | 0.00001 | $45.89 \pm 1.50$ | $26160.53 \pm 1253.68$ |
|  | 0.0001 | $52.24 \pm 1.60$ | $19088.42 \pm 809.65$ |
|  | 0.001 | $58.25 \pm 1.43$ | $14744.07 \pm 560.55$ |
|  | 0.01 | $\mathbf{60.61} \pm 1.60$ | $14896.93 \pm 296.23$ |
|  | 0.1 | $54.16 \pm 0.78$ | $19904.50 \pm 187.42$ |

Table 2: Performance of the noisy Dirichlet model on CIFAR-10, used to plot results in Figure 3(a). We show one standard deviation over 3 trials.

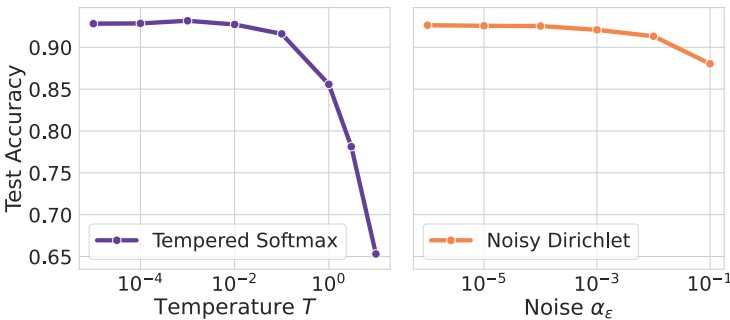

Figure 16: Test accuracy on corrupted CIFAR-10 for both tempered softmax (left) and noisy Dirichlet. Noisy Dirichlet performs very similar to the best performing softmax temperatures ($10^{-3}$) across a range of $\alpha_\epsilon$ values.

## K.6 Model Calibration

Section 4 is in general related to miscalibration. We, however, note that in standard neural networks, miscalibration typically means *overconfidence*, while in our work we show that Bayesian neural networks are typically *underconfident*. Both issues are related to model misspecification, where the model does not correctly represent the data-generating process. We show that we can address part of the model misspecification by explicitly encoding our assumptions about aleatoric uncertainty with likelihood tempering or the noisy Dirichlet model.

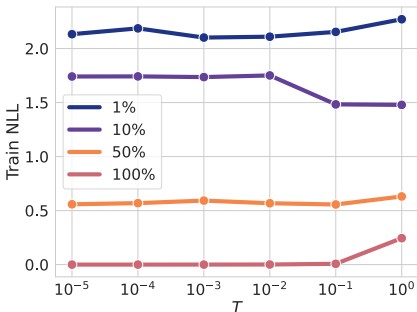

Figure 17: For various subsets of the unaugmented CIFAR-10 dataset, we visualize the average sample train NLL as a measure of the train data fit using ResNet-18. We find that increasing the cardinality of the dataset improves the fit over the training data. Conclusively, cardinality of the dataset cannot explain the underfitting as observed in Figure 5.

In Table 3, we present the test expected calibration error (ECE) on CIFAR-10 with 20% label noise for the model with standard softmax likelihood, tempered softmax likelihood and the noisy Dirichlet model. The noisy Dirichlet model provides the best calibration (i.e. lowest ECE), and suggests a more accurate description of the data-generating process.

| Model | Test BMA Accuracy (%) | Test ECE |
|---|---|---|
| Noisy Dirichlet ($\alpha_\epsilon^\star = 0.01$) | **88.68** | **0.024** |
| Tempered Softmax ($T^\star = 0.1$) | 88.45 | 0.144 |
| Standard Softmax | 80.62 | 0.179 |

Table 3: On CIFAR-10 with 20% label noise, we compare the test Expected Calibration Error (ECE) between three models — noisy Dirichlet with optimal noise parameter $\alpha_\epsilon^\star$ (see Table 1), tempered softmax with optimal temperature $T^\star$ (see Table 2), and standard softmax. We find that the noisy Dirichlet model, performs at least as well than the tempered softmax model, while being significantly better calibrated. In general, we expect models which more accurately represent the data-generating process to be better calibrated.

## L   Experimental Details

We use PyTorch for all experiments [50]. For the Gaussian process experiments, we used GPyTorch [15] for all but the Laplace bernoulli classifier where we used scikit-learn [51].

We used $\mathcal{N}(0, 1)$ priors unless otherwise stated and ran cyclical learning rate SGLD for 1000 epochs with an initial learning rate of $1e - 6$ and a momentum term of 0.99. We used ResNet-18 [20] architectures unless otherwise stated and used the CIFAR-10[8].

For experiment management, we used wandb [5] to manage experiments. Experiments were performed on a mix of Nvidia RTX GPUs on internal servers and clusters as well as some RTX GPUs on AWS; we also used some AMD mi50 GPUs on an internal cluster. Each SGLD trial took roughly 10 hours to run, with a total of 3 trials per experiment. Over the course of all experiments including ones that did not make it into the paper, this is approximately 1 GPU year. Unless otherwise mentioned, all experiments used 3 random seeds and we plotted the mean with shading via one standard deviation.

### L.1   Synthetic problem

We present the code used to generate the data for the synthetic problem considered in Appendix E in Listing 1 following Maddox et al. [39].

---

[8] https://www.cs.toronto.edu/~kriz/cifar.html dataset

For all models we use an iid Gaussian prior $\mathcal{N}(0, 0.3^2)$ over the parameters. We use HMC with a step size $3 \cdot 10^{-6}$ and the trajectory length is $\pi \cdot 0.3/2$, amounting to $150 \cdot 10^3$ leapfrog steps per iteration, following the advice in Izmailov et al. [31]. We run HMC for 100 iterations, discarding the first 10 samples as burn-in.

For data augmentation, we generate $K = 4$ augmentations of each datapoint. We run HMC on the distribution in Eq. (11). For the tempered likelihood, we use $T = 0.1$, and for the noisy Dirichlet model we use $\alpha_\epsilon = 10^{-5}$.

Listing 1: Data generation for the synthetic problem

```python
import numpy as np

def twospirals(n_samples, noise=.5, random_state=920):
    """
     Returns the two spirals dataset.
    """
    onp.random.seed(random_state)

    n = np.sqrt(np.random.rand(n_samples,1)) * 600 * (2*np.pi)/360
    d1x = -1.5*np.cos(n)*n + np.random.randn(n_samples,1) * noise
    d1y =  1.5*np.sin(n)*n + np.random.randn(n_samples,1) * noise
    return (np.vstack((np.hstack((d1x,d1y)), np.hstack((-d1x,-d1y)))),
            np.hstack((np.zeros(n_samples), np.ones(n_samples))))

x, y = twospirals(n_samples=200, noise=0.6, random_state=920)

label_mask = (np.linalg.norm(x, axis=-1) > 13)
y[label_mask] = 1 - y[label_mask]

mask = np.logical_and(x[:, 0] < 0, x[:, 1] < 0)
x, y = x[mask], y[mask]
x_aug = np.concatenate([x, -x,
                        x * onp.array([[1., -1]]),
                        x * onp.array([[-1., 1]])])
y_aug = np.concatenate([y, y, y, y])
```