# OpenReview forum: "On Uncertainty, Tempering, and Data Augmentation in Bayesian Classification"
_NeurIPS.cc/2022/Conference — NeurIPS 2022 Accept_

### Official Review · Reviewer_fpwD · 2022-07-01

**Rating:** 7
**Confidence:** 5
**Soundness:** 3 good
**Presentation:** 2 fair
**Contribution:** 3 good

**Summary:**

The authors discuss the relationship between tempering and data augmentation and build a Dirichlet distribution-based approach that allows them to explain this relationship and how to use it in a more principled manner.


**Questions:**

- Can the authors comment on why the finding on the softened likelihood is 'counterintuitive' (l57)? I would have assumed that data augmentation, giving us essentially more noisy covariates, would unsurprisingly lead to a less confident model.

**Limitations:**

Limitations and broader impact have been discussed. However, only in the appendix.

**Strengths And Weaknesses:**

## Strengths
- The task is well motivated and discussed.
- The proposed solution gives a clear and structured approach to the relationship between tempering and augmentation.

## Weaknesses
- The writing has several weaknesses, not in the style, where it is well written, but rather in the formulations and presentation which make common knowledge sound too special and new. E.g. l28 makes it sound, especially with the typographical emphasis on 'accuracy' as if the dependency of the accuracy on the aleatoric uncertainty were a surprise. If we tell our models not to trust the data too much, i.e., assume high aleatoric uncertainty, then of course this will lead the GP towards a smoother fit, changing its predictions as well. The same goes for the scenario in Figure 1(c). If we know that we have different assumptions about noise in the data and therefore the smoothness of our decision boundary we would and should incorporate that in our BNN by stronger/weaker regularization instead of using the same likelihood and prior.
Another example is the paragraph around l150. Here again, the finding is accompanied by an emphasis on 'training data'. But if we have a strong prior with a weak probability of observing any specific class we will still have a rather weak posterior after only a single observation. This is not a surprising statement worth an exclamation mark. Repeated around l170, where a different prior gives us different confidence in the posterior. Similarly to the $\alpha_\epsilon$ approach around eq (5), this is not a surprise, but simply the statement that our prior influences our posterior.
Note: I do see why the authors use these examples and similar parts as building blocks to develop their storyline. They would just read a lot better if presented less as surprises and more as _here are some well-known facts about our Bayesian models. Now let us tell you, dear reader, how they motivate the approach we propose_.
- A major weakness is the complete lack of any discussion on prior work using Dirichlet distributions as part of a deep (Bayesian) classification model. Malinin et al. (2018) are cited, but only with respect to the uncertainty decomposition. A discussion of this work and follow-up work (Malinin et al., 2019) is missing. Similarly, Sensoy et al. (2018), Charpentier et al. (2020), etc. all tackle the deficiencies of relying on a simple softmax-based categorical likelihood by introducing Dirichlet priors in different ways.


## Other
- Fig 2: Please show the full training data, not just the unaugmented one either here or in the appendix, for the reader to get a clear picture of what data the model actually sees during training.
- l253: Technically we have not only not received new information in _our_ reality. In the model's reality, it has received new information through these repetitions of the same observations, leading to the behaviour you correctly describe.
- Figure 3 and Fig 5 (a) lack the error bars of the remaining plots.
- Similarly, the number of runs the error bars in the experiments correspond to are currently only available in the checklist and the appendix. They should be mentioned somewhere in the main paper as well (beginning of Sec 6 and/or figure caption).

---

> ### Author Response · Authors · 2022-08-02
> **Author Response to Reviewer fpwD**
>
> We appreciate you recognizing the clarity of our approach, and thank you for your thoughtful comments.
>
>
> ## Dirichlet-based Uncertainty Models
>
> Thank you for the relevant pointers. We have added an extended discussion of these works to Section 2 of the updated draft. The methods in this family modify the standard neural network model to represent both aleatoric uncertainty and epistemic uncertainty with a single model, and are often proposed as an efficient alternative to deep ensembles and Bayesian neural networks. These methods are typically evaluated on uncertainty calibration, anomaly detection and adversarial robustness. We, on the other hand, use the standard Bayesian neural network model with categorical observations, and study the Dirichlet distributions over the class probabilities as a tool to reason about the representation of aleatoric uncertainty, rather than as a modification of the model. Our focus is Bayesian classification in general, with emphasis on predictive performance and results on the cold posterior effect, data augmentation and label noise.
>
> We also note that we do cite the Dirichlet observation model, and that we make many significant contributions that are unrelated to this model, which we have highlighted in the general response. In particular, we provide the first general explanation of the cold posterior effect (covering numerous isolated examples from other works), and its direct link to data augmentation. We also make the important observation that Bayesian neural network classifiers are significantly underrepresenting aleatoric uncertainty.
>
> ## Writing style
>
> We want to clarify that we are not intending to say it is surprising that aleatoric uncertainty affects predictions. Rather, we believe it is truly notable that deep Bayesian classifiers often represent the same assumptions about aleatoric uncertainty, despite the fact that our beliefs about aleatoric uncertainty should vary significantly across datasets and models, and profoundly affect our predictions. We also don’t think that there is a particular suggestion of surprise in some of these excerpts, such as l170, where we are simply stating the outcome of a calculation. We do think it is remarkable that Bayesian classifiers are significantly underrepresenting aleatoric uncertainty on the training data: here our confidence in the class labels should be near 100% for the benchmarks being considered, for instance, in the cold posteriors paper, but with standard observation models it is significantly less. This observation is particularly pertinent in the context that many efforts to model label noise are trying to accommodate for increased rather than decreased aleatoric uncertainty. In any case, we will clarify the emphasis, and appreciate the feedback.
>
> ## Data Augmentation and Softened Likelihood
>
> > why the finding on the softened likelihood is 'counterintuitive' (l57)?
>
> We believe that this finding may be counter-intuitive to some readers due to the fact that typically in Bayesian learning the posterior contracts as the number of observations increases. In a way, data augmentation is related to increasing the size of the dataset, but has the opposite effect. Indeed, in the appendix Figure 17 (Appendix J.5), we show that increasing the size of the dataset leads to a better fit to the training data due to posterior contraction; with the data augmentation, however, we do not get the same contraction, and increasing the strength of augmentation leads to a lower-quality fit to the training data, as we show in Figure 5.
>
> While we believe that this result may appear counter-intuitive, we provide a detailed explanation of the mechanism behind it in Section 5. See also our response to reviewer JCLG for further discussion of this point.
>
> ## Other Questions
>
>
> The mean and standard deviation has been computed over 3 runs. We present the complete results table for CIFAR-10 in our response to Reviewer JCLG under the heading “Detailed results on robustness to hyper-parameters”. The error bars appear missing in the plots due to their magnitude being relatively small in Figures 3 and 5, especially for the NLL plots where the y-axis is logarithmically scaled. As you recommended, we have made explicit our choice of mean and standard deviation over 3 trials in Section 6, and added a results table to Appendix J.2 (Tables 1 and 2).
>
> ## Conclusion
>
> Thank you again for your thoughtful review. We have done our best to accommodate your questions, and hope you can consider increasing your score in light of these responses. Please feel welcome to ask us if you have any further questions, and we will respond if the system permits.

---

> > ### Comment · Reviewer_fpwD · 2022-08-03
> > **Answer to parts of the rebuttal**
> >
> > Thank you for your detailed answer.
> >
> > > Dirichlet-based Uncertainty Models
> >
> > I agree that the proposed approach goes beyond what the cited previous authors have been doing in their approaches. My complaint was primarily about the fact that the existence of previous work on using Dirichlet distributions (+softmax or on their own) as part of the pipeline was completely missing from the original submission. The new paragraph in the related work solves this complaint entirely. It acknowledges that the use of Dirichlet has been widely explored in recent years, but makes it clear that while this paper also uses this distribution and knows about the prior work, it goes in a different direction with it.
> >
> > > We want to clarify that we are not intending to say it is surprising that aleatoric uncertainty affects predictions. Rather, we believe it is truly notable that deep Bayesian classifiers often represent the same assumptions about aleatoric uncertainty, despite the fact that our beliefs about aleatoric uncertainty should vary significantly across datasets and models, and profoundly affect our predictions.
> >
> > I wholeheartedly agree. My reading was probably too critical. It read to me like a text rediscovering that our assumptions actually matter and be included properly in our models rather than just be based on computational convenience as we do too often. Instead of it being a reminder of this fact.
> > But I see how one can read it less critically and indeed get to the second interpretation of the text. (My prior that recently every paper tries to rediscover the basics might have been too strong ;) )
> >
> > >  We also don’t think that there is a particular suggestion of surprise in some of these excerpts, such as l170, where we are simply stating the outcome of a calculation.
> >
> > l170 (now l177), does indeed just give the outcome of the calculation. But its partner, l150 (now l157) is written as "...confident in the correct label on the _training data_!". This does suggest some surprise instead of being, as in l177, a simple outcome of a calculation essentially just repeating what the GP example in Figure 1 already demonstrated. A strong prior against a specific outcome, whether implied or explicitly stated, dominates the observed result. But again, this is primarily about the style. I agree with the conclusion you draw from the fact that this implied prior can be too strong and should be properly accounted for.

---

> > > ### Author Response · Authors · 2022-08-03
> > > **Thanks for your reply reviewer fpwD. Could you please consider updating your score in the original review?**
> > >
> > > Thanks for your thoughtful reply. We're glad your questions are addressed and that we seem to be on the same page. Thanks again also for sending the pointers. We really enjoyed reading those papers, and feel our paper is now stronger with the added discussion. Would you please consider updating the score in your original review, to reflect your current evaluation given our response and updated paper?

---

### Official Review · Reviewer_1PhB · 2022-07-03

**Rating:** 6
**Confidence:** 4
**Soundness:** 3 good
**Presentation:** 4 excellent
**Contribution:** 2 fair

**Summary:**

This work investigates the impact of explicitly modeling aleatoric uncertainty in Bayesian Neural Networks (BNNs). They demonstrate empirically and theoretically that it equates to tempering the likelihood function and link it to the recently observed cold posterior effect under data augmentation. Moreover, this work proposes to model aleatoric uncertainty by parameterizing the models output using a noisy Dirichtlet distribution in order to eradicate the cold posterior effect for data augmentation in BNNs.

**Questions:**

- Can the authors comment on why they equate input-agnostic tempering of the likelihood with modeling aleatoric uncertainty appropriately? I would expect that aleatoric uncertainty is an input dependent quantity.
- Can the authors comment on W1. I would consider voting for acceptance if this concern is adequately addressed.
- The effect observed in this work seems to be connected to the more general phenomenon that NNs are poorly calibrated (over/under confident) [1]. Do the authors think that the observation for BNNs (i.e. cold posterior effect for data augmentation) is simply a symptom of this more general problem?


**Strengths And Weaknesses:**

Strengths:
- The theoretical link between data augmentation and the cold posterior effect is very interesting, although entirely this papers novelty
- The effect of modeling aleatoric uncertainty on BNNs is an underexplored area
 - This work provides an empirical investigation of great depth

Weaknesses:
- W1 I am not entirely convinced by the finding that the noisy Dirichlet parameterization eliminates the necessity for tempering the likelihood (Fig. 4). If I am not mistaken the noise parameter in the noisy Dirichlet parameterization is equivalent to changing the temperature of the resulting model. Thus, for a fairer comparison in Fig. 4 the authors would have to plot the test accuracy of their proposed method over a larger range of Ts. With the given range of Ts it cannot be judged whether the drawn conclusions are true. In particular, these experiments are conducted on CIFAR10 which does not contain aleatoric uncertainty as mentioned by the authors. Thus by setting the noise parameter very small the authors implicitly offset their temperature towards lower values.
- I think the usage of the term aleatoric uncertainty in this work is confusing. The general trend seems to be that aleatoric uncertainty is a quantity that depends on the input data which seems to be not the case in this work.

---

> ### Author Response · Authors · 2022-08-02
> **Author Response to Reviewer 1PhB (Part 1 / 2)**
>
> Thank you for your thoughtful comments and recognizing the depth of our investigation.
>
> We want to emphasize that our paper makes many timely and novel contributions, and is relevant to researchers interested in aleatoric uncertainty in general, and the effects of data augmentation on Bayesian classifiers, and not only researchers wanting to explain the cold posterior effect (although that is also a significant contribution). We have highlighted some of these contributions in the general response, but reiterate them here:
> We show the standard softmax likelihood is insufficient to accurately represent beliefs about aleatoric uncertainty.
> We show cold posteriors are a way to specify aleatoric uncertainty
> We show the precise way in which data augmentation leads to underconfidence in the likelihood, which resolves the empirical connection between data augmentation and cold posteriors.
> For the first time, we show that we can remove the cold-posterior effect with a valid Bayesian model by more accurately accounting for aleatoric uncertainty via the noisy Dirichlet model.
>
> We address your questions and concerns in detail below.
>
> ## W1: Tempering and Noisy Dirichlet Model
>
> We would like to clarify that the noisy Dirichlet parameterization is **not** equivalent to tempering the likelihood. See, for example, the form of the tempered posterior in Eq. (4) and the form of the noisy Dirichlet model in Eq. (5). The two distributions are not equivalent to each other, and we cannot recover the tempered posterior by varying $\alpha_\epsilon$ in the noisy Dirichlet model, and vice versa.
>
> Moreover, as we explain in detail in the response to Reviewer stWW, the noisy Dirichlet model corresponds to a valid Bayesian model with a correct prior and likelihood. Tempered likelihood posteriors do not have the same interpretation.
>
> **Increased range of $T$s.** Following your suggestion, we run additional experiments on CIFAR-10 with tempered softmax likelihood, where we include lower temperatures $T$ of 1e-6, 1e-7 and 1e-8. The mean test BMA accuracy and NLL is noted in the table below, alongside old results for comparison.
>
> | Temperature $T$     | Test BMA Accuracy (%) | Test NLL |
> | ------ | ------------------ | ------------ |
> | 1e-8 | 93.15              | 3536.67  |
> | 1e-7 | 93.01              | 3568.50  |
> | 1e-6 | 93.13              | 3569.81  |
> | 1e-5 | 92.97              | 3577.74  |
> | 1e-4 | 93.13              | 3599.74  |
> | 1e-3 | **93.28**        | **3476.33**  |
> | 1e-2 | 92.94              | 3521.37  |
> | 1e-1 | 91.77              | 3587.20  |
>
> In comparison, the noisy Dirichlet model at $\alpha_\epsilon = 10^{-6}$ achieves $93.28\%$ test BMA accuracy, and performs relatively similarly at different temperatures.
>
> We reiterate that the most important observation in Figure 4 is that additional tempering does not change the overall behavior of the noisy Dirichlet likelihood, while making a significant impact for the softmax likelihood. We argue that the drop in performance at T = 1 for the softmax likelihood is the consequence of inadequate representation of aleatoric uncertainty. Finally, we do not imply that one model is universally better than the other, but only argue that tempering or the noisy Dirichlet model both allow for a more accurate representation of our beliefs about the aleatoric uncertainty in the data-generating process. With the noisy Dirichlet model, we have explicit control over the aleatoric uncertainty and can achieve strong results with no need for additional tempering.
>
> We hope that we addressed your concerns regarding W1, and you will consider raising your score. Please feel welcome to ask us if you have any further questions, and we will respond if the system permits.

---

> > ### Author Response · Authors · 2022-08-02
> > **Author Response to Reviewer 1PhB (Part 2 / 2)**
> >
> > ## Aleatoric Uncertainty
> >
> > In general, aleatoric uncertainty is the irreducible uncertainty present in the data. In this work, we focus on classification problems where the aleatoric uncertainty corresponds to label noise (incorrectly labeled examples). We explore the assumptions about the amount of label noise encoded in Bayesian models, and how these assumptions are affected by likelihood tempering and data augmentation. We also propose the noisy Dirichlet model, which allows us to more explicitly control the assumptions about label noise encoded in the model.
> >
> > **Input dependence.** We note that our discussion of aleatoric uncertainty does not assume that label noise is input-independent. For example, a Bayesian neural network model can be much more confident in its predictions on some examples compared to others. The aleatoric uncertainty is input-dependent both with standard softmax likelihood, with tempering, and with the noisy Dirichlet model. However, the temperature parameter in the tempered softmax likelihood models, and the $\alpha_\epsilon$ parameter in the noisy Dirichlet model, allow us to increase or decrease the amount of aleatoric uncertainty globally, with a single scalar parameter.
> >
> > The reason that a single parameter is sufficient is that a priori we have no knowledge of the aleatoric uncertainty for individual data points. Generally, we can consider a noisy Dirichlet model with input-dependent noise parameter $\alpha_\epsilon$. A model like that would be justified if our data came from multiple sources with different amounts of label noise, e.g. if we combined two datasets together. We can then naturally define a different value of $\alpha_\epsilon$ for different data points. Exploring input-dependent noisy Dirichlet models (and input-dependent tempering) is an exciting direction for future work.
> >
> > ## Connection to Calibration
> >
> > In general, our observations are indeed related to miscalibration. However, we note that in standard neural networks, miscalibration typically means *overconfidence*, while in our work we show that Bayesian neural networks are typically *underconfident*. Both issues are related to model misspecification, where the model does not correctly represent the data-generating process. In this work, we show that we can partially address model misspecification by explicitly encoding our assumptions about aleatoric uncertainty with likelihood tempering or the noisy Dirichlet model.
> >
> > Inspired by your question, we evaluate the test expected calibration error (ECE) on CIFAR-10 with 20% label noise for the model with standard softmax likelihood, tempered softmax likelihood and the noisy Dirichlet model. The noisy Dirichlet model provides the best calibration (i.e. lowest ECE).
> >
> > | Model     					         | Test BMA Accuracy (%) | Test ECE |
> > | --------------------------------------------------------------- | ------------------ | ------------ |
> > | Noisy Dirichlet ($\alpha_\epsilon^\star$ = 1e-2)  | **88.68**              |  **0.024**  |
> > | Tempered Softmax ($T^\star$ = 1e-1) 	         | 88.45       	| 0.144  |
> > | Softmax ($T$ = 1) 				         | 80.62              | 0.179  |
> >
> > In general, we expect models which more accurately represent the data-generating process to be better calibrated.
> >
> > ## Conclusion
> >
> > Thank you for your questions and suggestions! We hope that we addressed your concerns, and you will consider raising your score. We will be happy to address any further questions if the system permits.

---

> > > ### Comment · Reviewer_1PhB · 2022-08-03
> > > **Re: Author Response to Reviewer 1PhB (Part 2 / 2)**
> > >
> > > I thank the authors for the detailed response. My concerns have been adequately addressed and I am voting for acceptance.

---

> > > > ### Author Response · Authors · 2022-08-03
> > > > **Thanks for your response reviewer 1PhB --- could you please consider updating your score?**
> > > >
> > > > Thanks for your response, and your thoughtful review. Could you please consider updating the score in your original review to reflect that your concerns are now addressed?

---

### Official Review · Reviewer_JCLG · 2022-07-08

**Rating:** 7
**Confidence:** 3
**Soundness:** 4 excellent
**Presentation:** 4 excellent
**Contribution:** 4 excellent

**Summary:**

This work closely examines aleatoric uncertainty in Bayesian neural networks for classification. It demonstrates how tempered softmax (cross-entropy likelihood), standard softmax and their proposed method, the Dirichlet observation model, behave under varying degrees of aleatoric uncertainties. They show that via the Dirichlet model or by tempering the likelihood function, the prior belief about aleatoric uncertainty can be expressed in BNNs for classification. In this way, the authors provide an important understanding of why BNNs fit poorly to the training data when using data augmentation, as well as tools for addressing the issue.

**Questions:**

Regarding the results in Fig. 3. In lines 305-306: "We plot the results for the best performing ... $T$ ... or ... $\alpha_\epsilon$...". To me, the effect of varying these values wrt the results in Fig. 3 is clearly missing. It would help the community and/or practitioners a lot to know more about the robustness of these parameters wrt to the label noise. E.g. for CIFAR's BMA test accuracy, $\alpha_\epsilon=1e-04$ is the best performing hyperparam when label noise = 0.0, while $\alpha_\epsilon=1e-02$ is the way to go when labe noise = 0.2. This is a big jump.

1. What happened at label noise = 0.2 when $\alpha_\epsilon \in [1e-04, 1e-02)$? Or for larger values? If you tested $T\in[10^-5, 10]$ and $\alpha_\epsilon\in[10^-6,10^-1]$, why not include these? A clarification of this would be very helpful and interesting.

2. Smaller comment on Fig. 3. What noise values and temperatures did you use to produce the intermediate points? Such as label noise = 0.5 in Fig. 3a? Please explicitize.

Regarding the paragraph "Data augmentation leads to underfitting on train data."

3. It seems unsurprising that a model that does not use data augmention is more overfitted to the train data than one that uses no augmentation. Can't the underfitting on the training data in part be explained by data augmentation giving more samples for the model to fit? In other words, to which extent would you say that the reduced quality of the fit on the original training data is due to the increased cardinality of the augmented dataset ($\mathcal{D}'$), and not the softened likelihood?



**Limitations:**

Yes; found in the appendix. Above I have addressed a small concern about the transparency of the robustness of the tempered softmax and Dirichlet model wrt their parameters. Constructive suggestions for improvement are given above.

**Strengths And Weaknesses:**

***Strengths***

The paper is exemplarily **clearly** written and is easy to follow. The useful explanation of how to combine tempering with data augmention between lines 246-256 is an example of the pedagogical nature of the work.

Experiments are conducted in order to back up claims in the text. For instance, the augmentation vs. tempering experiment neatly supports the ideas in the paragraph between lines 246-256. Also, I find the paper to be technically sound. Hence the **quality** is good. I have some thoughts on improvement outlined below and in the "Questions" field.

The Dirichlet observation model appears **original**, but especially the insights regarding the effect of data augmentation in BNNs are novel and should be impactful. The related work seems OK and carefully written.

***Weaknesses***

**Quality:** Limitations of the proposed method and the tempered softmax can be more elaborated in Fig. 3. See Questions for more explicit feedback.

**Clarity:** You are referencing Fig. 4 (line 287) before you reference Fig. 3. (line 303). You should be able to switch the order of the two corresponding paragraphs as they seem to be the same length.

In the current form of the work, I find the use of the word "honest" when referring to prior beliefs about aleatoric uncertainty slightly awkward as I am unsure what is meant by it. I suggest to add an explanation, for instance in lines 8-11 "... misrepresenting our honest beliefs about aleatoric uncertainty, as [insert explanation of how the honest belief is misrepresented]" or elsewhere. Alternatively the term could be dropped as it is not being used much in the later parts of the work.

Typos:
* line 9: we show [that] data augmentation
* missing punctuation in bold-fonted title in line 328.
* the sentence in lines 185-187 could be less tediously written. For instance "However, Wenzel et al. [51] show, the tempered" --> "However, Wenzel et al. [51] show that the tempered".

---

> ### Author Response · Authors · 2022-08-02
> **Author Response to Reviewer JCLG (Part 1 / 3)**
>
> Thank you for your highly thoughtful and supportive comments. We appreciate that you found the paper instructive and high quality, recognizing the generality and impact of our contributions.
>
> ## Detailed Results on Robustness to Hyperparameters
>
> We address your request and report the BMA Test Accuracy and Test NLL for all combinations of label noise and temperature in the tempered softmax model or $\alpha_\epsilon$ parameter in the noisy Dirichlet model for CIFAR-10.
>
> Tempered softmax likelihood:
> | Label Noise | Temperature $T$     | Test BMA Accuracy (%) | Test NLL |
> | ----------- | ------- | --------------------- | -------- |
> | 0           | 0.00001 | 92.97 $\pm$ 0.19               | 3577.74 $\pm$ 36.69  |
> |             | 0.0001  | 93.13 $\pm$ 0.08                 | 3599.74 $\pm$ 14.91  |
> |             | 0.001   | **93.28**  $\pm$ 0.08               | 3476.33 $\pm$ 18.85  |
> |             | 0.01    | 92.94 $\pm$ 0.12                | 3521.37 $\pm$ 32.88 |
> |             | 0.1     | 91.77 $\pm$ 0.26                | 3587.2 $\pm$ 56.83  |
> |             | 1       | 85.78 $\pm$ 0.19                 | 5344.59 $\pm$ 31.25  |
> |             | 3       | 78.24 $\pm$ 0.13                 | 7891.12 $\pm$ 48.70  |
> |             | 10      | 65.69 $\pm$ 0.36                 | 12440.25 $\pm$ 61.04 |
> | 0.1         | 0.00001 | 88.43 $\pm$ 0.08                 | 5872.89 $\pm$ 104.96  |
> |             | 0.0001  | 88.15 $\pm$ 0.23                 | 6000.17 $\pm$ 124.29  |
> |             | 0.001   | 88.49 $\pm$ 0.35                | 5885.82 $\pm$ 33.58  |
> |             | 0.01    | 88.23 $\pm$ 0.33                 | 5777.52 $\pm$ 97.05  |
> |             | 0.1     | **89.71** $\pm$ 0.20               | 4322.14 $\pm$ 35.02  |
> |             | 1       | 83.08 $\pm$ 0.44                 | 6419.33 $\pm$ 69.41  |
> |             | 3       | 74.51 $\pm$ 0.15                 | 9088.32 $\pm$ 35.38  |
> |             | 10      | 60.7 $\pm$ 1.01                 | 13551.9 $\pm$ 81.70 |
> | 0.2         | 0.00001 | 83.3 $\pm$ 0.37                  | 8346.63 $\pm$ 87.89  |
> |             | 0.0001  | 83.21 $\pm$ 0.40                 | 8425.63 $\pm$ 185.47 |
> |             | 0.001   | 82.97 $\pm$ 0.57                | 8345.62 $\pm$ 284.03  |
> |             | 0.01    | 83.49 $\pm$ 0.09                 | 7920.99 $\pm$ 103.56  |
> |             | 0.1     | **88.45** $\pm$ 0.61                 | 5448.62 $\pm$ 125.64 |
> |             | 1       | 80.62 $\pm$ 0.21                 | 7774.44 $\pm$ 27.49  |
> |             | 3       | 71.97 $\pm$ 0.46                 | 10406.1 $\pm$ 57.89  |
> |             | 10      | 54.85 $\pm$ 1.37                 | 15327.49 $\pm$ 295.44 |
> | 0.5         | 0.00001 | 60.31 $\pm$ 1.10                 | 17986.78 $\pm$ 364.34 |
> |             | 0.0001  | 58.87 $\pm$ 0.65                 | 18381.33 $\pm$ 187.42  |
> |             | 0.001   | 59.60 $\pm$ 0.33                  | 17816.36 $\pm$ 258.46 |
> |             | 0.01    | 62.40 $\pm$ 0.43                 | 15499.53 $\pm$ 168.51 |
> |             | 0.1     | **81.91** $\pm$ 0.39                 | 9998.57 $\pm$ 185.99  |
> |             | 1       | 68.70 $\pm$ 0.55                 | 13187.17 $\pm$ 73.68 |
> |             | 3       | 56.54 $\pm$ 0.40                | 16126.59 $\pm$ 78.99 |
> |             | 10      | 41.85 $\pm$ 0.54                | 19707.41 $\pm$ 91.98 |
> | 0.8         | 0.00001 | 27.96 $\pm$ 1.22                | 29862.19 $\pm$ 1241.84 |
> |             | 0.0001  | 27.43 $\pm$ 1.36                 | 31027.27 $\pm$ 382.76 |
> |             | 0.001   | 28.08 $\pm$ 0.65                 | 29515.91 $\pm$ 637.08 |
> |             | 0.01    | 33.56 $\pm$ 1.43                 | 22806.11 $\pm$ 1126.21 |
> |             | 0.1     | **59.90** $\pm$ 0.45                  | 17524.06 $\pm$ 81.75 |
> |             | 1       | 41.23 $\pm$ 0.36                 | 20839.3 $\pm$ 58.07  |
> |             | 3       | 34.44 $\pm$ 1.29                | 21811.33 $\pm$ 68.24 |
> |             | 10      | 27.96 $\pm$ 1.20                 | 23129.04 $\pm$ 21.34 |

---

> > ### Author Response · Authors · 2022-08-02
> > **Author Response to Reviewer JCLG (Part 2 / 3)**
> >
> > Noisy Dirichlet model:
> > | Label Noise | Noise $\alpha_\epsilon$ | Test BMA Accuracy (%) | Test NLL |
> > | ----------- | -------------------- | --------------------- | -------- |
> > | 0.          | 0.000001             | 92.75 $\pm$ 0.24                | 8080.56 $\pm$ 131.65  |
> > |             | 0.00001              | 92.72 $\pm$ 0.10                | 6874.20 $\pm$ 36.90  |
> > |             | 0.0001               | **92.78**  $\pm$ 0.09               | 5586.79 $\pm$ 86.25  |
> > |             | 0.001                | 92.15 $\pm$ 0.07                | 4491.19 $\pm$ 117.73  |
> > |             | 0.01                 | 91.50 $\pm$ 0.07                 | 3564.69 $\pm$ 50.87  |
> > |             | 0.1                  | 88.26  $\pm$ 0.07               | 6384.59 $\pm$ 15.93  |
> > | 0.1         | 0.000001             | 89.11 $\pm$ 0.29                 | 11507.79 $\pm$ 345.42 |
> > |             | 0.00001              | 89.11 $\pm$ 0.24                 | 9638.78 $\pm$ 324.75  |
> > |             | 0.0001               | 89.70 $\pm$ 0.06               | 7399.10 $\pm$ 64.58  |
> > |             | 0.001                | **90.28** $\pm$ 0.37                | 5195.21 $\pm$ 48.68  |
> > |             | 0.01                 | 90.25 $\pm$ 0.39                | 3955.32 $\pm$ 77.80 |
> > |             | 0.1                  | 86.93 $\pm$ 0.05               | 7194.98 $\pm$ 24.86  |
> > | 0.2         | 0.000001             | 85.36 $\pm$ 0.33                | 14221.11 $\pm$ 161.47 |
> > |             | 0.00001              | 85.72 $\pm$ 0.40                | 11884.17 $\pm$ 241.67 |
> > |             | 0.0001               | 86.71 $\pm$ 0.24                | 8865.18 $\pm$ 187.91  |
> > |             | 0.001                | 88.09 $\pm$ 0.18                | 6006.20 $\pm$ 83.82 |
> > |             | 0.01                 | **88.68** $\pm$ 0.22                | 4471.03 $\pm$ 62.10  |
> > |             | 0.1                  | 85.48 $\pm$ 0.01                | 8144.23 $\pm$ 11.36 |
> > | 0.5         | 0.000001             | 73.80 $\pm$ 1.12                | 19231.71 $\pm$ 760.60 |
> > |             | 0.00001              | 74.54 $\pm$0.99                | 16510.63 $\pm$ 755.69 |
> > |             | 0.0001               | 76.85 $\pm$ 0.33                | 12659.80 $\pm$ 540.86 |
> > |             | 0.001                | 79.86 $\pm$ 0.29                | 8708.74 $\pm$ 136.45 |
> > |             | 0.01                 | **82.56** $\pm$ 0.19                | 6995.20 $\pm$ 54.81  |
> > |             | 0.1                  | 77.55 $\pm$ 0.54                | 12331.45 $\pm$ 26.44 |
> > | 0.8         | 0.000001             | 43.71 $\pm$ 2.93                | 31334.21 $\pm$ 2914.70 |
> > |             | 0.00001              | 45.89 $\pm$ 1.50                | 26160.53 $\pm$ 1253.68 |
> > |             | 0.0001               | 52.24 $\pm$ 1.60                | 19088.42 $\pm$ 809.65 |
> > |             | 0.001                | 58.25 $\pm$ 1.43                | 14744.07 $\pm$ 560.55 |
> > |             | 0.01                 | **60.61** $\pm$ 1.60                | 14896.93 $\pm$ 296.23 |
> > |             | 0.1                  | 53.55 $\pm$ 0.78                 | 19904.50 $\pm$ 187.42 |
> >
> > Both with the temperature value in the tempered softmax likelihood model, and with the noise parameter $\alpha_\epsilon$ in the noisy Dirichlet model, we control the assumptions about the level of label noise encoded in the model. The results are consistent with the intuition presented in the paper: no one model can perform optimally across all levels of label noise, and we need to specify our assumptions through $T$ or $\alpha_\epsilon$ to achieve best results. However, we also note that the noisy Dirichlet model appears relatively robust to the choice of $\alpha_\epsilon$, with reasonable performance across a range of $\alpha_\epsilon$ values across the board.
> >
> > We include these results in the **Appendix J.2** of the updated paper. In **Figure 3** of the main text, we show the best results for each label noise level to avoid visual clutter.

---

> > > ### Author Response · Authors · 2022-08-02
> > > **Author Response to Reviewer JCLG (Part 3 / 3)**
> > >
> > > ## Data Augmentation and Increased Data Cardinality
> > >
> > > Inspired by Q3, we provide an experiment where we compare the effect of data augmentation on the aleatoric uncertainty to the effect of increasing the size of the dataset. In Figure 5 of the paper, we show that the model with data augmentation achieves higher NLL on the training data, meaning that the model with no augmentation fits the data better. Next, below we compare the BMA NLL on the training data for models trained on different dataset sizes:
> > >
> > > | Subset | $T$ | Train Accuracy          | Avg. Train NLL       |
> > > |--------|-------------|--------------------|------------------------|
> > > | 1%   | 1e-06       | 0.2140              | 2.1566              |
> > > | 1%   | 1e-05       | 0.2080              | 2.1330              |
> > > | 1%   | 0.0001      | 0.1840              | 2.1869     |
> > > | 1%   | 0.001       | 0.2220              | 2.1018     |
> > > | 1%   | 0.01        | 0.2340              | 2.1098              |
> > > | 1%   | 0.1         | 0.2220              | 2.1543              |
> > > | 1%   | 1.0         | 0.1340              | 2.2713                |
> > > | 10%    | 1e-06       | 0.5694             | 1.6872             |
> > > | 10%    | 1e-05       | 0.5496             | 1.7413     |
> > > | 10%    | 0.0001      | 0.5498             | 1.7423     |
> > > | 10%    | 0.001       | 0.5516             | 1.7359     |
> > > | 10%    | 0.01        | 0.5446             | 1.7511             |
> > > | 10%    | 0.1         | 0.6012             | 1.4833             |
> > > | 10%   | 1.0         | 0.4980              | 1.4785             |
> > > | 50%    | 1e-06       | 0.8746            | 0.5340            |
> > > | 50%    | 1e-05       | 0.8679            | 0.5583     |
> > > | 50%    | 0.0001      | 0.8672            | 0.5690            |
> > > | 50%    | 0.001       | 0.8672            | 0.5925            |
> > > | 50%    | 0.01        | 0.8814             | 0.5680     |
> > > | 50%    | 0.1         | 0.8940            | 0.5565            |
> > > | 50%    | 1.0         | 0.8568            | 0.6308            |
> > > | 100%    | 1e-05       | 1.0                | 3.2612e-05  |
> > > | 100%    | 0.0001      | 1.0                | 3.7123e-05  |
> > > | 100%    | 0.001       | 1.0                | 0.0001 |
> > > | 100%    | 0.01        | 1.0                | 0.0008  |
> > > | 100%    | 0.1         | 1.0                | 0.0073   |
> > > | 100%   | 1.0         | 0.9747 | 0.2447    |
> > >
> > > We report these results visually in the **appendix Figure 17 (Appendix J.5)**. Here, for each run we use the specified subset of the CIFAR-10 training data, and produce 50 samples from the posterior with SGLD at different temperatures. We then evaluate the BMA accuracy and NLL on the training dataset. We observe the *opposite* result to our data augmentation experiments. Increasing the data size leads to a better fit to the training data!
> > >
> > > **Explanation of the results.** The difference between the results for the dataset size and data augmentation can be understood through posterior contraction. When we increase the number of iid samples, the posterior contracts according to the likelihood. For example, if a single observation $(x, y)$ is repeated $N$ times in the dataset $D = {(x, y), (x, y), \ldots, (x, y)}$, the posterior will be $p(w | D) \propto p(w) \cdot p(y \vert x, w)^N $; as $N$ grows, the model will become more and more confident in the label $y$ on the input $x$. However, with data augmentation, the posterior does not contract according to the number of observed data points. Consequently, increasing the number of augmentations makes the fit worse and not better. This mechanism is explained in detail in Section 5 of the paper.
> > >
> > > To sum, the effect of data augmentation cannot be explained as simply adding more data. We include this experiment in Figure 17 (Appendix J.5) of the updated paper, which we believe helps clarify the difference between augmentations and dataset cardinality in how they affect aleatoric uncertainty. Many thanks for your question.
> > >
> > > ## Conclusion
> > >
> > > We again really appreciate your supportive remarks. We have put a significant effort into adding experiments to help address your questions, and updated the paper based on your feedback. We hope you can consider increasing your score in light of our response. We are happy to answer any further questions if the submission system permits.

---

> > > > ### Comment · Reviewer_JCLG · 2022-08-04
> > > > **Comments on rebuttal**
> > > >
> > > > Thank you for the serious efforts made in addressing my concerns. I am content with the response and will consider raising the score during the discussion period with the other reviewers and the AC.

---

### Official Review · Reviewer_stWW · 2022-07-12

**Rating:** 7
**Confidence:** 4
**Soundness:** 3 good
**Presentation:** 2 fair
**Contribution:** 3 good

**Summary:**

The authors propose a novel strategy for representing aleatoric uncertainty in classification problems. They connect their method to the use of cold posteriors and data augmentation, and empirically demonstrate that it provides an effective alternative to tempering.

**Questions:**

Where does the representation of aleatoric uncertainty come from? Is there a well-defined posterior over some quantity that corresponds to aleatoric uncertainty?

How do you see the relationship between this work and the existing literature on Dirichlet-based uncertainty for modeling aleatoric and epistemic uncertainty jointly (see e.g. Kopetzki, …, Gunnemann, Evaluating Robustness of Predictive Uncertainty: Are Dirichlet-Based Models Reliable?, 2020 and the work cited therein in the third paragraph)

**Limitations:**

It’s unclear what effects the method has on uncertainty quantification, a key use of Bayesian inference in general.

**Strengths And Weaknesses:**

The paper offers a nice, clearly-explained overview of the cold posterior effect and its possible interpretations. It also offers a nice explanation of its relationship to data augmentation.

The method itself, however, I cannot understand as a Bayesian procedure, a serious weakness in a paper that seems to aim to provide a “valid” Bayesian method for representing uncertainty. In particular, in the proposed method, the normalizing constant of the likelihood p(y | f(x)) will in general depend on f(x) (since sum_{y = 1}^C p(y | f(x)) does not necessarily equal 1 when using the likelihoods in equations 4 and 5). Therefore, the proportionality given in line 164 — the central justification for the method, as a tempered likelihood approach — does not seem to hold. (Note that in general, the reason why tempered likelihoods are difficult to justify from a purely Bayesian perspective is because they directly imply that the likelihood does not sum to 1; ideas such as coarsening or adding an unobserved class are used to get around this problem).

Also, the use of proportionality notation everywhere is generally very confusing. I’d appreciate a clear statement of the generative process behind the model somewhere, including its prior and likelihood. Lines 135-139 are extremely confusing as they make it seem as though a Dirichlet prior is being introduced into the model, when in fact the authors are only constructing a likelihood function in a particular form (if I understand correctly).

---- UPDATE ----

The authors have explained their proposed method more clearly in our discussion. With the understanding that they will rewrite the section of the paper introducing the proposed method to increase the clarity substantially, and that they will add discussion on (a) the softmax temperature prior approach and (b) previous work on modeling aleatoric uncertainty in classifiers with Dirichlet distributions, I am happy to raise my score. I sincerely thank the authors for their committed engagement, and am optimistic that once explained concisely and clearly the ideas will be a valuable contribution to the literature.

---

> ### Author Response · Authors · 2022-08-02
> **Author Response to Reviewer stWW (Part 1 / 2)**
>
> Thank you for your thoughtful questions, which we address below. We want to highlight that our method is correct. There seem to have been some misunderstandings, which we have clarified.
>
> ## Correctness of the Method
>
> **Tempered likelihood is not a valid likelihood.** We want to clarify that we are not arguing that likelihood tempering corresponds to a likelihood that sums up to 1 over the possible values of the observed variable: indeed, as you mentioned,
> $\sum_y f_y(x, w)^{1/T} \ne 1$. In fact, we make this point in the paragraph “Is a Tempered Likelihood a Valid Likelihood?”, and in particular lines 190-192.
>
> **Noisy Dirichlet model is a valid Bayesian model.** However, the Noisy Dirichlet model described in Section 4.3 is different from the tempered likelihood model, and corresponds to a valid likelihood. In fact, the final posterior in Eq. (6) can be written as
> $p_{ND}(w \vert D) \propto p(w) \cdot \prod_{x, y \in D} \text{Dir.}(\alpha_{\epsilon}, \ldots, \alpha_{\epsilon})(f(x, w)) \cdot f_y(x, w)$.
> Here $f_y(x, w)$, the only term that depends on the observed class labels, is just the standard softmax likelihood, and in particular $\sum_{y} f_y(x, w) = 1$ for all $x$.
> The term $q_{ND}(w) \propto p(w) \cdot \prod_{x, y \in D} \text{Dir.}(\alpha_{\epsilon}, \ldots, \alpha_{\epsilon})(f(x, w))$ should be interpreted as a prior over the parameters.
>
> **$q_{ND}$ is a valid distribution over w.** First, the $q_{ND}(w)$ is a valid distribution over $w$: for a fixed $\alpha_{\epsilon}$ the Dirichlet density is bounded, so the term $\prod_{x, y \in D} \text{Dir.}(\alpha_{\epsilon}, \ldots, \alpha_{\epsilon})(f(x, w))$ is bounded by a constant $B$ as a function of $w$. Consequently, $p(w) \cdot \prod_{x, y \in D} \text{Dir.}(\alpha_{\epsilon}, \ldots, \alpha_{\epsilon})(f(x, w))$ is a product of a probability density $p(w)$ and a bounded function, so $\int_{w} p(w) \cdot \prod_{x, y \in D} \text{Dir.}(\alpha_{\epsilon}, \ldots, \alpha_{\epsilon})(f(x, w)) dw  = Z < B$.  Consequently, $q_{ND}(w) = \frac 1 {Z} p(w) \cdot \prod_{x, y \in D} \text{Dir.}(\alpha_{\epsilon}, \ldots, \alpha_{\epsilon})(f(x, w))$ is a valid probability density over $w$.
>
> **$q_{ND}$ is a valid prior over w.** Finally, we argue that $q_{ND}(w)$ can be used as a valid prior over the parameters of the neural network. We make this point in lines 207-213 of the updated paper. Indeed, while the prior $q_{ND}$ depends on the training data inputs $x$, it does not depend on the training labels $y$. The Bayesian neural network defines a probabilistic model for the labels $y$ and does not model the inputs $x$, so the standard issues with data-dependent priors or empirical Bayes do not apply to $q_{ND}$. See, for example, the EmpCov prior in [1] .
>
> **Summary.** In the paper, we are not trying to argue that standard tempering corresponds to a valid likelihood, which is not the case, as you mentioned. However, we explain that it provides a useful mechanism for characterizing aleatoric uncertainty in the data, which is lacking in standard BNNs. With the noisy Dirichlet model, we propose a valid Bayesian procedure which achieves similar results to tempering, but with a valid likelihood (standard softmax likelihood). We will clarify this point further.

---

> > ### Author Response · Authors · 2022-08-02
> > **Author Response to Reviewer stWW (Part 2 / 2)**
> >
> > ## The Proportionality Notation
> >
> > Thank you for your comment, we clarify the use of the proportionality notation in the paper. In general $g(w) \propto f(w)$ means that the left hand side of the equation is equal to the right hand side up to a scalar multiplier $Z$ (which does not depend on $w$), i.e. $g(w) = Z \cdot f(w)$ for all values of $w$.
> >
> > For example, in lines 141-145 we say $f_y(x) \propto \text{Dir.}(1, \ldots, 1)(f(x)) \cdot f_y(x)$. Here, the proportionality is with respect to the class probability vector $f(x)$, and holds trivially, as $\text{Dir.}(1, \ldots, 1)(f(x)) = (C-1)!$, where $C$ is the number of classes, is a scalar constant that does not depend on $f(x)$. This transition does not change the model, or introduce a new prior, it is simply an algebraic derivation that is useful for interpreting the posterior distribution over the class probabilities, as we do in Sections 4.1 to 4.3.
> >
> > Based on your feedback, we updated the paper to explicitly state which variable the proportionality is with respect to, and to explain the meaning of proportionality notation more clearly.
> >
> > **A clear statement of the generative process.** The data generative process corresponding to the standard Bayesian neural network is described by the following distribution:
> > $p(Y) = \int_{w} p(Y | w) p(w) dw$, where $Y$ represents all the class labels of the training data points, and $w$ represents the weights of the model. For BNNs with conventional priors, we can further specify the conditioning on the input features: $p(Y | X) = \int_{w} p(Y | X, w) p(w) dw$, where the likelihood $p(Y | X, w)$ is the iid softmax likelihood described in line 120 of the paper. For the noisy Dirchlet model, the conditioning is more general: $p(Y | X) = \int_{w} p(Y | X, w) p(w | X) dw$, where the likelihood  $p(Y | X, w)$ is the iid softmax likelihood (same as before), and the prior $p(w | X)$ is the $q_{ND}$ prior defined in Eq. (6).
> >
> > As you mentioned in your review, and as we discussed above, the tempered likelihood model does not strictly correspond to any data generation process.
> >
> > ## Other Questions
> >
> > > Where does the representation of aleatoric uncertainty come from? Is there a well-defined posterior over some quantity that corresponds to aleatoric uncertainty?
> >
> > In the context of classification, the aleatoric uncertainty corresponds to the amount of label noise. By the representation of aleatoric uncertainty, we refer to the confidence of the model in its predictions, e.g. on the training data. In the paper, we consider the expected confidence of the model in the observed class labels on the training data as a qualitative measure corresponding to its representation of aleatoric uncertainty (lines 153-157, 173-177, 201-205).
> >
> > > How do you see the relationship between this work and the existing literature on Dirichlet-based uncertainty for modeling aleatoric and epistemic uncertainty jointly
> >
> > Thank you for the relevant pointers, we added a discussion of the Dirichlet-based uncertainty models to the related work in Section 2. The methods in this family modify the standard neural network model to represent both aleatoric uncertainty and the epistemic uncertainty with a single model, and are often proposed as an efficient alternative to deep ensembles and Bayesian neural networks. These methods are typically evaluated on uncertainty calibration, anomaly detection and adversarial robustness. We, on the other hand, use the standard Bayesian neural network model with categorical observations, and study the Dirichlet distributions over the class probabilities as a tool to reason about the representation of aleatoric uncertainty, rather than as a modification of the model. Our focus is Bayesian classification in general, with emphasis on (i) predictive performance especially under aleatoric uncertainty due to label noise, (ii) understanding the cold posterior effect, and (iii) interaction of data augmentation with aleatoric uncertainty.
> >
> > ## Conclusion
> >
> > We thank you again for these questions. We have updated the paper accordingly, to include clarifications. We emphasize again that the method we use is in fact correct.
> >
> > We also want to note, as in our general response, that the paper makes several significant and timely contributions that go beyond explaining cold posteriors, and the use of the Dirichlet observation model, such as explaining the connection between aleatoric uncertainty in SGLD and data augmentation, and the observation that Bayesian classifiers in general are typically underrepresenting aleatoric uncertainty in standard benchmarks.
> >
> > **References**
> >
> > [1] Dangers of Bayesian Model Averaging under Covariate Shift;
> > Pavel Izmailov, Patrick Nicholson, Sanae Lotfi, Andrew Gordon Wilson

---

> > > ### Comment · Reviewer_stWW · 2022-08-05
> > > **Still some confusion...**
> > >
> > > Thanks for this thorough and thoughtful response. I see that my previous understanding of what the paper is proposing was incorrect; unfortunately, I'm still not 100% sure I understand what the paper is actually proposing.
> > >
> > > Here's my \textit{guess} at what the method is, though I still can't reconcile it with everything you've said, so perhaps you can clarify further. Fundamentally, you are proposing that when using Bayesian neural network classifiers, one should consider \textit{tilting} one's initial priors. In particular, you propose a tilting function that takes the form of a Dirichlet pdf, with tunable concentration parameter; this is a convenient way of upweighting or downweighting conditional probabilities over the outcome variable that are sparse (high confidence) or flat (low confidence).
> > >
> > > First: is this understanding correct?
> > >
> > > Second: whether or not this understanding is correct, I have serious concerns over the clarity of this paper. While I understand that this may be a failing on my part, I hope (having spent many years studying theoretical Bayesian statistics and Bayesian machine learning, and many hours studying this paper to try to understand what's going on) that the authors will take the clarity issue seriously. To my mind (provided my current guess as to what's going on is correct) a clear statement of the motivation and method would be:
> > > 1. Priors on Bayesian neural networks are difficult to specify but important. An especially important feature is the amount of probability mass they put on conditional distributions $p_w(y|x)$ that have low versus high aleatoric uncertainty (corresponding to low versus high conditional entropy $H(y \mid x)$).
> > > 2. Motivated by the cold posterior effect and related work, one should consider tilting one's initial prior on the BNN weights to put more weight on conditional distributions with low aleatoric uncertainty. In particular, starting from the standard model:
> > > $W \sim p(w)$ and $Y \sim p_W(Y \mid X)$, one should consider a modified model with $W \sim \frac{1}{Z} g(w \mid X) p(w)$, where $g(w \mid X)$ is a tilting function that depends on the input training data and $Z$ is the normalizing constant.
> > > 3. In particular, you propose $g(w \mid X) = Dir(\alpha)( p_W(Y \mid X))$, which encourages the prior to put large mass on functions that have low aleatoric uncertainty at the input/training data.
> > > In particular, I think this kind of explanation at the beginning would make it really clear (a) what the basic problem is (specifying a prior), (b) what the original model is, and (c) how precisely you modify the model to address the problem.
> > >
> > > Third, I have many questions and concerns about this approach (as I understand it)
> > > 1. A standard way of controlling aleatoric uncertainty in Bayesian classifiers is by controlling the prior on the temperature of the softmax. In particular, if the output probabilities are given by applying a softmax function to continuous-valued inputs, you multiply those inputs by a temperature parameter with a prior on low values before taking the softmax. To my mind, this softmax temperature approach accomplishes essentially the same goals as the method the authors propose, but has two distinct advantages: (1) it does not result in a prior with unknown normalizing constant, (2) it ensures that the prior puts weight on low uncertainty functions on the test data (and, indeed, the entire input space), not just the training datapoints.
> > > 2. More generally, what is the intended effect of the method on points besides the training data?
> > > 3. The approach is a way of \textit{modifying} an initial prior. If the initial prior already has low aleatoric uncertainty it presumably should not be lowered further. It seems important to obtain better control.
> > > 4. Typically, asymptotically, the effects of the prior will wash away. Here, you have a data-dependent prior, so the large-data asymptotics are unclear. Is this method intended to affect the aleatoric uncertainty even in the large data limit?
> > > 5. It would be helpful to the reader to clarify the relationship of the method to the large literature on tilting, and in particular its relationship to exponential tilting.

---

> > > > ### Author Response · Authors · 2022-08-07
> > > > **Clarifications for stWW (Part 1/2)**
> > > >
> > > > Thank you for engaging with our rebuttal and for your thoughtful feedback. We are happy to address any questions or concerns and provide further clarifications.
> > > >
> > > > Your summary of the noisy Dirichlet model is correct. We modify the prior over the network parameters by forming a Dirichlet distribution over the predicted class probabilities on the training data points (see Eq. 3 for the precise expression). We also want to note that although our exchange has revolved around this model, it is one of many contributions in the paper, primarily serving a focused purpose: a demonstration that with a different prior, which more accurately accommodates our beliefs about aleatoric uncertainty, there is no cold posterior effect, even with data augmentation.
> > > >
> > > > Our paper is more broadly about the aleatoric uncertainty in Bayesian classification. In particular, we analyze the effect of posterior / likelihood tempering and data augmentation on the aleatoric uncertainty in Bayesian neural network classifiers. The noisy Dirichlet model arises as a natural alternative to the likelihood tempering from our analysis in Sections 4.1-4.3 of the paper.
> > > >
> > > > We appreciate your feedback, and will incorporate your suggestions into the presentation where the noisy Dirichlet model is introduced, in Section 4.3.
> > > >
> > > > We now answer your questions about the noisy Dirichlet model:
> > > >
> > > > **Question 1: Why not just use softmax temperature?** If we understand correctly, the softmax temperature approach that you mention is using the likelihood in the form $p_T(y \vert x, w) = \text{softmax}(l_1(x, w) / T, \dots, l_C(x, w) / T)[y]$, where $y$ is the class label, and $l_c(x, w)$ are the logits corresponding to the class $c$ produced by the network with weights $w$ on input $x$. In fact, we consider this modification of the model in detail in the appendix E of the paper. We refer to it as the *smoothed softmax likelihood model*. We show that smoothed softmax is insufficient to model the aleatoric uncertainty in Bayesian classification. In particular, in Figure 10 we show that the smoothed softmax likelihood does not address the cold posterior effect.
> > > >
> > > > Intuitively, the smoothed softmax likelihood corresponds to a simple reparameterization of the standard softmax likelihood BNN, where the last layer parameters are all divided by a scalar constant $T$. This model is equivalent to simply rescaling the prior over the last layer parameters. In particular, for BNNs with zero-mean iid Gaussian priors, this reparameterization is equivalent to dividing the standard deviation of the prior over the last layer parameters by $T$. Such rescaling has a very limited effect on the properties of the model, especially given the homogeneity of the ReLU activations and the scale invariance due to normalization layers, which may provide further intuition for why the smoothed softmax likelihood is insufficient for modeling aleatoric uncertainty in Bayesian classification.
> > > >
> > > > **Question 2: What is the intended effect on points beyond the training data?** The prior that we specify on the training data defines the types of solutions that we expect, and has a significant effect on the predictions on the test data. See, for example, Figure 1 in our paper. Our assumptions about the aleatoric uncertainty have a significant effect on the predictions that the model makes and the types of solutions that we find. The model in Figure 1 (a) makes very different predictions from the model on Figure 1 (b), even though they only differ by the specified noise variance on the training data. Similarly, for classification, the model with the decision boundary shown in blue in Figure 1(c) makes very different predictions from the model with decision boundary shown in red, and we can distinguish between them based on the assumptions on aleatoric uncertainty on the training data.
> > > >
> > > > Empirically, our experiments, in particular Figures 3, 4, 5, show that the prior in the noisy Dirichlet model has a significant impact on the predictions on the test datapoints, which we can see from the difference in accuracy. For example, in Figure 4, the standard sotmax ($T=1$) model only achieves 86% accuracy on CIFAR-10, while the noisy Dirichlet model achieves 93% test accuracy.

---

> > > > > ### Author Response · Authors · 2022-08-07
> > > > > **Clarifications for stWW (Part 2/2)**
> > > > >
> > > > > **Question 3: What if the aleatoric uncertainty is too low?** While our main motivation is the scenario where the aleatoric uncertainty is too high, we can use the same ideas for the situation when the aleatoric uncertainty is too low. We can achieve this effect by using $\alpha_\epsilon > 1$.
> > > > >
> > > > > Note that in Section 6.3 we apply the BNNs with the noisy Dirichlet model priors to datasets with high label noise, and achieve strong results. Interestingly, even for these datasets, the standard BNNs appear to overestimate the aleatoric uncertainty, and the optimal noise parameter $\alpha_\epsilon$ (or temperature $T$ in the tempered softmax model) are below 1.
> > > > >
> > > > > **Question 4: Asymptotic behavior in the large data limit.** For practical Bayesian neural networks and realistic datasets, we want the prior to have an effect on the solutions that we find, as demonstrated by our experiments: the noisy Dirichlet model consistently achieves better results than the standard $T=1$ softmax model. Our prior will continue to have an effect as the size of the data grows. If desired, in the large data regime, it is easy to modify the model to have a prior of fixed strength as the size of the data grows. But the current prior suits our purpose of showing how we can represent aleatoric uncertainty to eliminate the need for tempering.
> > > > >
> > > > > **Question 5: Relationship to tilting.** Our prior in the noisy Dirichlet model can be written as
> > > > > $q_{ND}(w) \propto_w p(w) \cdot \prod_{i=1}^n \text{Dirichlet}(\alpha_\epsilon, \ldots, \alpha_\epsilon)(f(x_i, w))$, where $f(x_i, w)$ is the vector of predicted class probabilities on input $x_i$ produced by the network with weights $w$. Generally, we can view this transformation as some form of tilting of the prior distribution, in the sense that we multiply the prior density by a function of $w$, and we will add remarks to this effect in the paper.
> > > > >
> > > > > We do not believe that our specific transformation is a case of *exponential* tilting, however, as the Dirichlet distribution density cannot be represented in the form $\text{Dirichlet}(\alpha_\epsilon, \ldots, \alpha_\epsilon) (f(x_i, w)) \propto e^{\theta^T T(w)}$ for some sufficient statistic $T$. The Dirichlet distribution is an exponential family, so it can be represented as $h(w)  e^{\theta^T T(w) - A(\theta)}$, but the base measure $h(w)$ is non-trivial, so $q_{ND}(w)$ is not an exponential tilting of $p(w)$.

---

> > > > > > ### Comment · Reviewer_stWW · 2022-08-07
> > > > > > **Clarifications**
> > > > > >
> > > > > > Thanks for your prompt and thoughtful response. Let me clarify two questions:
> > > > > >
> > > > > > **Question 1: Why not just use softmax temperature?** I understand of course that adjusting the softmax temperature is not equivalent to tempering, as discussed in Appendix E. But I thought you were proposing a method for adjusting the prior, not tempering? What I had in mind was the standard trick of placing a prior on the softmax temperature with large mass on low values. In particular, we can rewrite the initial model as $W \sim p(w)$, $T \sim \delta_1(t)$ and $Y \mid X \sim \mathrm{Categorical}(\mathrm{softmax}(f_1(X, W)/T, \ldots, f_C(X, W)/T)$. Then, to put more weight on low aleatoric uncertainty predictions we adjust the prior to e.g. $\log T \sim \mathrm{Normal}(\mu_T, 1)$, where $\mu_T$ is set to a large negative value. Indeed (provided the function class $f$ is bounded) the resulting prior over $p(Y \mid X)$ can put arbitrarily large weight on arbitrarily low aleatoric uncertainty solutions.
> > > > > >
> > > > > > **Question 2: What is the intended effect on points beyond the training data?** My concern here is that the prior you are proposing encourages low aleatoric uncertainty only on the *empirical distribution of the training data $X_1, \ldots, X_N$* as opposed to *the full support of the input space $\mathcal{X}$*. In other words, your proposed prior need not encourage low aleatoric uncertainty on unobserved points. (Technically, whether or not it does so depends crucially on the smoothness properties of the function class and the properties of the data generating distribution.) Putting a prior on the softmax temperature $T$, by contrast, can enforce low aleatoric uncertainty over the full input data space.
> > > > > >
> > > > > > One more comment, with respect to **Question 4: Asymptotic behavior in the large data limit.** Are you proposing that you do not want your model to learn the true aleatoric uncertainty even with infinite data? Usually in statistics we would want a regression model $p_w(Y| x)$ to converge to the true data-generating probability $p_0(Y \mid x)$ with infinite data. In Bayesian statistics, this is guaranteed (under weak regularity conditions) so long as the prior is fixed and the model is well-specified. The claim that you would want to increase the strength of the prior with more data such that the prior always has a non-negligible effect is somewhat radical, as there's an obvious danger of learning aleatoric uncertainty that is inaccurate and in particular too low. What's the justification here?

---

> > > > > > > ### Author Response · Authors · 2022-08-08
> > > > > > > **Updated Clarifications for Reviewer stWW**
> > > > > > >
> > > > > > > Thank you for your quick response.
> > > > > > >
> > > > > > > **Question 1: Why not just use softmax temperature?** Yes, the noisy Dirichlet model adjusts the prior and is not equivalent to tempering. However, in our experiments we show that the noisy Dirichlet model can successfully remove the cold posterior effect, and performs well on datasets with low label noise. On the other hand, the smoothed softmax model is unable to remove the cold posterior effect as illustrated in Figure 10 of the appendix. We summarize the results for ResNet-18 on CIFAR-10 in the table below.
> > > > > > >
> > > > > > > | Model                  | Best BMA test accuracy |
> > > > > > > |------------------------|------------------------|
> > > > > > > | $T=1$ softmax          | 85.8                   |
> > > > > > > | Tempered softmax       | 93.3                   |
> > > > > > > | Noisy Dirichlet model  | 92.8                   |
> > > > > > > | Smoothed softmax       | 87.6                   |
> > > > > > >
> > > > > > > Here for the tempered softmax, smoothed softmax and noisy Dirichlet model we report the best results across the values of $T$ and $\alpha_{\epsilon}$ respectively. The temperature $T$ in the smoothed softmax model does not provide sufficient control to specify our beliefs about aleatoric uncertainty.
> > > > > > >
> > > > > > > Our smoothed softmax model is a variation of the model that you describe, with the prior $T \sim \delta_{\hat{T}}(t)$, i.e. the prior is a delta function on some value of $T = \hat{T}$, which we vary in the experiments. We experimented with this model extensively, but did not achieve positive results. Since the model does not work well with a variety of values of $T$, in general we would not expect a Bayesian treatment of $T$ to be successful in our setting. We additionally explain our intuition for why the smoothed softmax model (Bayesian or not) performs poorly in our previous response. We are also unaware of any work successfully applying this model in the context of Bayesian neural network classification, but would be happy to include any relevant citations.
> > > > > > >
> > > > > > > **Question 2: What is the intended effect on points beyond the training data?** You are correct that the effect of our prior on the test datapoints will depend on the structure of the function class. However, the same can be said about the likelihood: we only evaluate the likelihood on the training data, so the effect on the predictions on the test datapoints will depend on the properties of the function class. Empirically (e.g. in the table above), we observe that for Bayesian neural networks the prior has a significant effect on the quality of predictions on the test data.
> > > > > > >
> > > > > > > Moreover, in lines 209-213 we discuss why we may not want to enforce low aleatoric uncertainty on all possible inputs. For example, CIFAR-10 images are carefully curated, and we want the model to make confident predictions on most of the training images. At the same time, for a white noise input image, we may not want the model to be confident in its predictions, as it does not correspond to any of the classes.
> > > > > > >
> > > > > > > **Question 4: Asymptotic behavior in the large data limit.** While we agree that in some cases we may want the effect of the prior to decrease as the size of the data grows, we disagree that our approach is radical. For example, in homoscedastic Gaussian process regression, the aleatoric uncertainty is specified through the noise variance parameter in the observation model. Consequently, the assumptions about the aleatoric uncertainty do not wash away, as the number of observed datapoints grows to infinity in these models.
> > > > > > >
> > > > > > > Furthermore, in this work, we are not specifically interested in the large data limit, we primarily focus on practical scenarios in the Bayesian neural network classification to make a very focused point about how tempering relates to our representation of aleatoric uncertainty. Thank you for your valuable comments; we will add a note on the large data limit regime where the noisy Dirichlet model is introduced, in Section 4.3.

---

> > > > > > > > ### Comment · Reviewer_stWW · 2022-08-09
> > > > > > > > **Thanks**
> > > > > > > >
> > > > > > > > Thanks for the clarifications. I'm not sure I agree with the comments on asymptotic behavior, but I understand this is somewhat outside the scope of the paper.
> > > > > > > >
> > > > > > > > I've updated my original review and raised the score. You have convinced me that this is a correct, nontrivial and interesting contribution. As discussed, I hope you'll take the issue of clarity especially seriously, and update the paper appropriately.

---

### Author Response · Authors · 2022-08-02
**General Author Response to all Readers (Part 1 / 2)**

We thank the reviewers for thoughtful and supportive feedback. We want to highlight that the paper is making many timely and significant contributions. In particular,
1. We show that standard classification likelihoods do not provide a mechanism to represent our beliefs about aleatoric uncertainty. A practitioner may use the same Bayesian neural network model on a dataset with high label noise, and on a benchmark dataset such as CIFAR-10, which has a very low degree of label noise.
2. We analyze the posterior distribution over the predictive class probabilities on the training data in order to understand the aleatoric uncertainty in classification and the effect of tempering and data augmentation.
3. We show that tempering with $T < 1$ increases the confidence of the model in the correct labels on the training data. On data with little or no label noise, such as e.g. CIFAR-10, the resulting posterior more accurately reflects our beliefs about the data compared to the $T=1$ model.
4. We explain the precise mechanism in which data augmentation, as conventionally applied in SGLD or other approximate inference procedures, softens the likelihood and leads to underconfidence in predictions. The effect of data augmentation is thus opposite to tempering. This counterintuitive result finally resolves the empirical connection between data augmentation and the cold posterior effect.
5. Based on our analysis, we propose the noisy Dirichlet model for BNN classification, which amounts to modifying the prior distribution over the parameters. The noisy Dirichlet model is a valid Bayesian model, with valid prior and likelihood. This model allows us to explicitly specify our beliefs about aleatoric uncertainty, leading to strong performance on a range of problems with varying label noise.
6. With a lognormal approximation to the noisy Dirichlet model, we for the first time remove the cold posterior effect in the presence of data augmentation.

Several papers published at NeurIPS and ICML are about observing the cold posterior effect in various instances, and hypothesizing why it might happen. We concretely explain these observations in great generality, which in itself is a major contribution. To supplement this explanation we also show how we can change the model to alleviate the cold posterior effect (the noisy Dirichlet model).

However, we note that our contributions also extend well beyond the cold posterior effect. While some works build models that account for label noise, the emphasis in these works is typically on overconfidence, and applications to datasets with significant label noise. Conversely, our paper is broadly about how Bayesian classifiers can represent too much aleatoric uncertainty. We explain how data augmentation leads to underconfidence with SGLD, and why Bayesian classifiers are severely underconfident on standard benchmarks, which are of much broader relevance than cold posteriors.

Inspired by reviewer comments, we have provided many clarifications and new experimental results. We summarize the new experimental results below:
* We provide the complete set of results over the full range of values $T \in [10^{-6}, 10]$ for the softmax likelihood and $\alpha_\epsilon \in [10^{-6}, 0.1]$ for the noisy Dirichlet model. The optimal hyperparameters remain consistent with our assumptions about the aleatoric uncertainty. Further, the noisy Dirichlet model appears relatively robust to the choice of $\alpha_\epsilon$. See our response to Reviewer JCLG for full details.
* We show that cardinality of the training dataset is not sufficient to explain underfitting due to data augmentation. To demonstrate this point, we construct unaugmented subsets of CIFAR-10 of different cardinality and compute the average training NLL. We find that increasing the number of data points in fact improves the training fit, while data augmentation makes the training fit worse. See our response to Reviewer JCLG for full details.
* We compute the test Expected Calibration Error (ECE) for the standard softmax model, a tempered softmax model, and the noisy Dirichlet model with the CIFAR-10 dataset at 20% label noise. We find that the noisy Dirichlet model provides the best ECE, agreeing with our intuition  that a more accurate representation of the data-generating process can not only improve model performance but model calibration as well. See our response to Reviewer 1PhB for full details.

---

> ### Author Response · Authors · 2022-08-02
> **General Author Response to all Readers (Part 2 / 2)**
>
> Next, we provide a list of updates in the revised version of our submission:
> * Tables 1 and 2 in the updated Appendix J.2 provide complete numerical results for all tested values of temperature for the softmax likelihood and noise parameter for the noisy Dirichlet model.
> * Figure 14 in the updated Appendix J.2 provides a visualization of the above numerical results for all tested values of temperature for the softmax likelihood and noise parameter for the noisy Dirichlet model.
> * Figure 17 in the updated Appendix J.5 provides a visualization of the train likelihoods for various subsets of the unaugmented CIFAR-10 dataset to show the effect of dataset cardinality on the data fit.
> * We add a discussion of the Dirichlet-Based Uncertainty (DUB) family of models to the related work in Section 2.
> * To clarify the proportionality notation, we specified which variable the proportionality is with respect to throughout the paper. We also added a footnote explaining the notation.
>
> We now respond to each reviewer individually.

---

### Meta-Review · Area_Chair_88LQ · 2022-08-27

**Recommendation:** Accept
**Confidence:** Less certain

**Metareview:**

This paper studies Bayesian neural networks and tempered posteriors and study the link between data augmentation and cold posteriors. Overall, after the rebuttal and discussion, all reviewers unanimously agreed to accept the paper. Reviewers appreciated the exploration of the link between data augmentation and the cold posterior effect and found the empirical results to be convincing.

One reviewer had concerns about the method but after discussion was convinced to accept the paper provided that the paper will be rewritten to increase clarity and that "they will add discussion on (a) the softmax temperature prior approach and (b) previous work on modeling aleatoric uncertainty in classifiers with Dirichlet distributions." Another reviewer stated that the discussion of limitations should be expanded in the paper. Please revise the paper to address the remaining concerns of the reviewers in the final version.

**Award:**

No

---

### Decision · Program_Chairs · 2022-09-14

Accept